# Holocene palaeoceanography of the Northeast Greenland shelf

Teodora Pados-Dibattista[1], Christof Pearce[1], Henrieka Detlef[1], Jørgen Brendtsen[2], Marit-Solveig Seidenkrantz[1]

[1]Paleoceanography and Paleoclimate Group, Arctic Research Centre, and iClimate Centre, Department of Geoscience, Aarhus University, Aarhus, 8000, Denmark
[2]ClimateLab, Brønshøj, 2700, Denmark

*Correspondence to*: Teodora Pados-Dibattista (pados.theo@gmx.at) and Marit-Solveig Seidenkrantz (mss@geo.au.dk)

**Abstract.** The Northeast Greenland shelf is highly sensitive to climate and ocean variability because it is swept by the East Greenland Current, which, through the western Fram Strait, forms the main pathway of export of sea ice and cold water masses from the Arctic Ocean into the North Atlantic Ocean. In order to reconstruct the variability of the East Greenland Current and general palaeoceanographic conditions in the area during the Holocene, we carried out benthic foraminiferal assemblage, stable isotope, and sedimentological analyses of a marine sediment core retrieved from the Northeast Greenland shelf (core DA17-NG-ST07-73G). The results reveal significant variations in the water masses and thus, in the strength of the East Greenland Current over the last ca. 9.4 ka BP. Between 9.4 and 8.2 ka BP the water column off Northeast Greenland was highly stratified, with cold, sea ice-loaded surface waters and strong influx of warm Atlantic Water in the subsurface. At ~8.4 ka BP a short-lived peak in terrestrial elements may be linked to influx of iceberg transported sediments and thus, to the so-called 8.2 ka event. Holocene Thermal Maximum like conditions prevailed from 8.2 to 6.2 ka BP, with a strong influence of the Return Atlantic Current and a weakened transport of Polar Water in the upper East Greenland Current. After 6.2 ka BP we recorded a return to a more stratified water column with sea ice-loaded surface waters and still Atlantic-sourced subsurface waters. After 4.2 ka BP increased Polar Water at the surface of the East Greenland Current and reduction of the Return Atlantic Water at subsurface levels signifies freshening and reduced stratification of the water column and (near) perennial sea-ice cover. The Neoglaciation started at 3.2 ka BP at our location, characterised by a strengthened East Greenland Current. Cold subsurface water conditions with possible sea-ice cover and minimum surface water productivity persisted here throughout the last ~3 kyr.

## 1 Introduction

The acceleration of climatic changes in the Arctic and Subarctic regions is particularly marked by the drastic reduction of summer sea-ice cover. According to model simulations, the Arctic Ocean may become seasonally ice-free as early as 2040-2050 (Stroeve et al., 2012). This sea-ice reduction has societal and environmental relevance, as it may open new opportunities for shipping and societal development of Arctic regions and Greenland. Furthermore, the sea-ice decline has been shown to alter Arctic ecosystems (Ardyna et al., 2014). However, despite their importance, Holocene marine environments and corresponding natural sea-ice states around Greenland are still not well understood. The Northeast (NE) Greenland continental shelf is a particularly important region for studying and understanding the mechanisms that control sea-ice formation, melt and drift. It is the broadest shelf along the Greenland margin, extending more than 300 km from the coastline. To the west it reaches the East Greenland coast with its major marine-terminating outlets of the Greenland Ice Sheet, while to the east it is bounded by the Fram Strait, which is the only deep passage between the Arctic Ocean and the rest of the world oceans. This shelf region is highly sensitive to climate and ocean variability because it underlies the East Greenland Current (EGC), which, through the western Fram Strait, is the main pathway of export of sea ice and cold water masses from the Arctic Ocean into the North Atlantic Ocean (Rudels and Quadfasel, 1991). The outflow of meltwater/sea ice is a key factor that determines the North Atlantic freshwater budget and stratification, and it influences

the deep-water formation. Moreover, waters on the NE Greenland continental shelf play a major role in the stability of the glacier outlets (Nioghalvfjerdsfjorden Glacier, Zachariæ Isstrøm, Storstrømmen) of the Northeast Greenland Ice Stream (Wilson and Straneo, 2015; Schaffer et al, 2017).

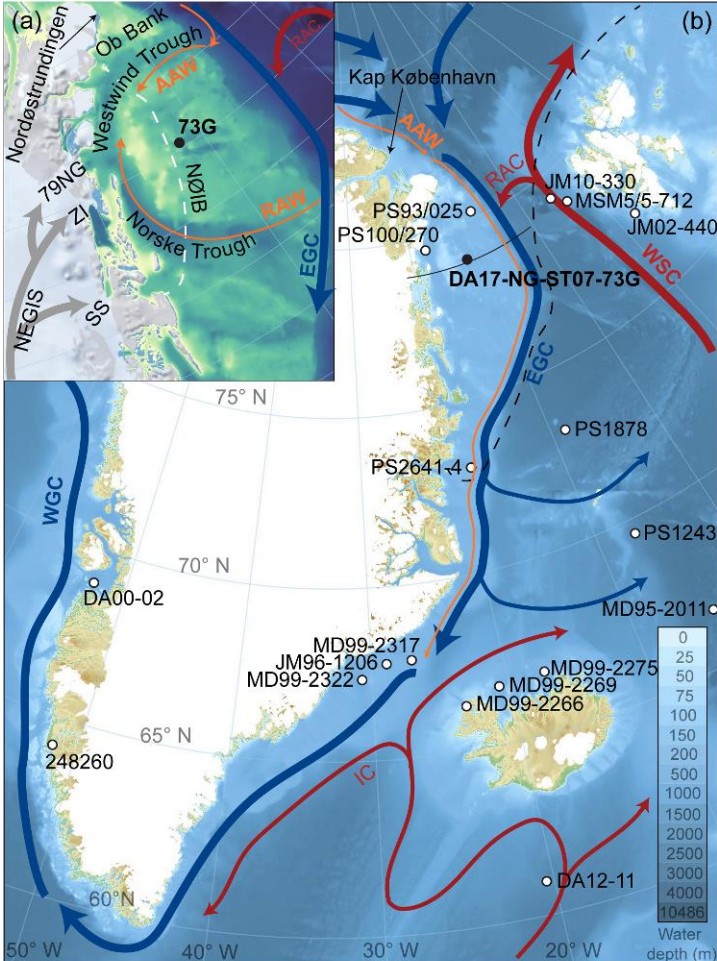

**Figure 1(a): Close-up of the Northeast Greenland shelf showing its characteristic features, the location of the studied sediment core (black circle) and the schematic subsurface circulation of the continental shelf (orange arrows). Arctic Atlantic Water (AAW), Return Atlantic Water (RAW), Nioghalvfjerdsfjorden Glacier (79NG), Zachariæ Isstrøm (ZI), Storstrømmen (SS), Northeast Greenland Ice Stream (NEGIS, grey arrows), Norske Øer Ice Barrier (NØIB; white dashed line). (b): Overview map of the study area with the studied sediment core DA17-NG-ST07-73G (black circle; 'Rumohr core DA17-NG-ST07-72R was taken from the same site), other core sites mentioned in the text (open circles) and schematic illustration of the major currents. East Greenland Current (EGC), West Spitsbergen Current (WSC), Return Atlantic Current (RAC), Irminger Current (IC), West Greenland Current (WGC). Red arrows: warm surface currents, blue arrows: cold surface currents, orange arrow: cooled, Atlantic originated subsurface water mass, Arctic Atlantic Water (AAW). Core sites: PS93/025 (Syring et al., 2020a; Zehnich et al., 2020), PS100/270 (Syring et al., 2020b), JM10-330 (Consolaro et al., 2018), MSM5/5-712 (Müller et al., 2012; Werner et al., 2013), JM02-440 (Ślubowska-Woldengen et al., 2007), PS1878 (Telesinski et al., 2014a-b), PS2641-4 (Müller et al., 2012; Perner et al., 2015; Kolling et al., 2017), PS1243 (Bauch et al., 2001), DA00-02 (Seidenkrantz et al., 2008), MD99-2317 (Jennings et al., 2011), JM96-1206 (Jennings et al., 2002; Perner et al., 2016), MD99-2322 (Jennings et al., 2011), MD99-2275 (Ran et al., 2006), MD95-2011 (Giraudeau et al., 2010), MD99-2269 (Giraudeau et al., 2004, Giraudeau et al., 2010), MD99-2266 (Moossen et al., 2015), 248260 (Seidenkrantz et al., 2007), DA12-11/2 (Van Nieuwenhove et al., 2018; Orme et al., 2018). Black line: location of the hydrographic transect pictured on Fig. 2b. Black dashed line: median sea-ice extent in September (1981-2010). The ocean bathymetry data are derived from GEBCO (Weatherall et al., 2015).**

At the surface, the EGC carries cold and fresh Polar Water (PW) southward. Below this relatively low-saline layer, the water masses originate from submerged and recirculated Atlantic-sourced waters from the Arctic (Arctic Atlantic Water, AAW), and partly from the Return Atlantic Current (RAC) (Quadfasel et al., 1987) (Fig. 1). The EGC transports these water masses from the north and along the East Greenland shelf towards the North Atlantic Ocean.

The freshwater budget here is one of the components that affects the strength of the Atlantic Meridional Overturning Circulation (AMOC; Rahmstorf, 1995; Clark et al., 2002). The AMOC transports warm and salty surface waters to high latitudes in the eastern Nordic Seas and the Arctic Ocean, where they cool, sink and return southwards at depth. Increased freshwater flux from the EGC
to the Nordic Seas and the northern North Atlantic may prevent deep convection, thus, reducing meridional heat transport, which causes cooling of the high latitudes (Clark et al., 2001). At present, due to the anthropogenic global warming, the AMOC may be in its weakest state in the last 1000 years (Caesar et al., 2021), demanding an improved understanding of this complex system and its components.

The amount of PWtransported southward in the EGC is related to ocean-atmosphere dynamics in the Arctic Ocean, such as the
70 Transpolar Drift (e.g., Mysak, 2001) and the regional wind stress above the Nordic Seas and Greenland shelf area; it thereby also depends on the North Atlantic Oscillation (NAO) and its Arctic counterpart, the Arctic Oscillation (AO) (e.g., Hurell et al., 2003). The NAO/AO is one of the most prominent and recurrent patterns of atmospheric circulation variability. The NAO/AO mode of variability refers to a redistribution of air masses between the Arctic and the subtropical Atlantic, linked to shifts in sea-level pressure. Its shifts from positive to negative phase produce large changes in the weather of the middle and high altitudes of the
75 Northern Hemisphere (Hurrell et al., 2003). In case of positive NAO/AO, strong south-westerlies carry moist air over Europe and Siberia, Atlantic Water (AW) inflow to the Arctic Ocean through the Fram Strait increases, and temperatures in the Arctic increase, which lead to a reduction in sea ice formation. During intervals of a negative NAO/AO these phenomena occur to be reversed (Kwok, 2000; Hurrell and Deser, 2010).

Several studies suggest that the East Greenland shelf has been subjected to a series of oceanographic and palaeoclimatic changes during the Holocene, induced by changes in the strength of the EGC linked to fluctuations in AW entrainment. Most of these studies focus on the shelf region of Middle and Southeast Greenland (e.g., Jennings et al., 2002; Jennings et al., 2011; Müller et al., 2012; Perner et al, 2015; Perner et al., 2016; Kolling et al., 2017) and only few studies have investigated the palaeoceanographic evolution of the EGC in the northern part of the East Greenland shelf (Bauch et al., 2001; Syring et al., 2020a, 2020b; Zehnich et
al., 2020).

In this study, we aim to reconstruct the dynamics and variability of the EGC off NE Greenland at centennial resolution throughout most of the Holocene, using micropalaeontological, sedimentological and geochemical analysis of a sediment core retrieved from the central NE Greenland shelf. Faunal assemblage analysis of benthic and planktic foraminifera, stable isotope measurements, radiocarbon datings and X-ray fluorescence data allow us to infer changes in sea surface productivity, subsurface temperatures,
sea-ice conditions and Greenland Ice Sheet melting over the last ca. 9.4 ka BP. The reconstruction provides new insights into the not yet resolved palaeoceanographic evolution of this region, with comparisons to published records from the larger polar/subpolar North Atlantic region.

## 2 Regional setting

The bathymetry of the NE Greenland continental shelf is characterised by a trough system, consisting of five prominent cross-shelf
troughs, which are deeper than the surrounding banks (Arndt et al., 2015). Nioghalvfjerdsfjorden Glacier (or 79 North Glacier), Zachariæ Isstrøm and Storstrømmen (Fig. 1) are the three major marine-terminating glaciers of the Northeast Greenland Ice Stream that drain in the area. Our study site is located approximately in the middle of the shelf in an inter-trough area between the Westwind Trough and Norske Trough, facing Nioghalvfjerdsfjorden Glacier and Zachariæ Isstøm (Fig. 1). The location lies directly on the flow path of the EGC, which flows southward along the Greenland shelf break (Johannessen, 1986) and over the middle to outer

shelf (Bourke et al., 1987). Below a layer of the relatively low-saline Polar Surface Water, mainly originating from the Arctic Ocean and with contribution from local melt water, the upper part of the EGC carries cold, low-saline (T: 0-1 °C, S<32) Polar Water (PW) from the Arctic Ocean in the upper ca. 200 m of the water column. At subsurface levels below the PW, the water mass has its main origin from Atlantic sources, i.e. to varying extent from the Return Atlantic Water (RAW; transported by the RAC) and from Arctic Atlantic Water (AAW; also sometimes named "Arctic Intermediate Water") (Rudels et al., 2005) (Fig. 2). The RAC is the western branch of the West Spitsbergen Current that transports relatively warm and saline water masses (T<2 °C, S: 34-35) from the North Atlantic towards NE Greenland and then, by joining the EGC, southwards along the Greenland coast. In contrast, the eastern branch of the West Spitsbergen Current transports AW into the Arctic Ocean, where it circulates at subsurface levels and cools. The AAW (T≥0 °C, S: 34-35) is formed by this recirculated cool AW, which again exits the Arctic Ocean and, together with the RAC, it forms the subsurface section of the EGC, as it flows southward along the East Greenland shelf (e.g., Quadfasel et al, 1987; Rudels, 2012). Along this southward path, the bathymetry on the NE Greenland shelf steers parts of the AW into the prominent cross-shelf troughs, toward the marine terminating glaciers of the Northeast Greenland Ice Stream, and thereby modulating the glaciers´ basal melt rates (Arndt et al., 2015; Schaffer et al., 2017).

The Polar Front in the Greenland Sea represents the eastern limit of perennial sea-ice cover. The location of the sea-ice edge in the summer depends on the extent of the sea-ice export from the Arctic Ocean (Vinje, 1977). Today the sea-ice margin and the location of the Polar Front lies east of our study site (Danish Meteorological Institute (DMI) and NSIDC, 2012). However, the Northeast Water Polynya is in fairly close proximity to our core location. This is a seasonally ice-free or only loosely ice-covered area south of Nordøstrundingen (Fig. 1) on the eastern Greenland coast. Two ice barriers support the seasonal formation of the Northeast Water Polynya. The shelf ice of the Ob Bank (Fig. 1a) pushes drift ice eastward, while the Norske Øer Ice Barrier (Fig. 1a) blocks the northward flowing sea ice, which is entrained in the NE Greenland coastal current (Schneider and Budéus, 1994). It starts to open around May/June, then gradually increases in size and closes in September (Pedersen et al., 1993). The polynya's maximum extent to the north can reach high latitudes up to 83°N, beyond the NE Greenland shelf. To the east, the polynya can occupy the entire NE Greenland shelf with only little sea ice left in the area (Schneider and Budéus, 1997).

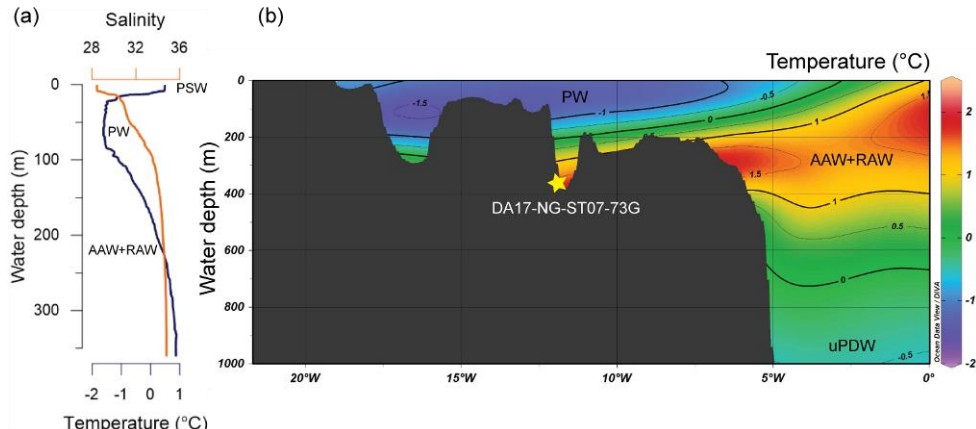

**Figure 2(a): Temperature and salinity profile of the water column at the coring station (79.073° N 11.918° W) obtained during the NorthGreen17 expedition, showing the main water masses. (b): Annual average temperature of the water column in the upper 1000 m along a transect at 78.876° N, showing the main water masses at the study site. Polar Surface Water (PSW), Polar Water (PW), combined Arctic Atlantic Water (AAW) and Return Atlantic Water (RAW), as well as upper Polar Deep Water (uPDW). The yellow star indicates the location of the core site at 385 m water depth. Temperature data from World Ocean Atlas 2018 (Boyer et al., 2018). The location of this transects is marked as a black line on Fig. 1b.**

## 3 Material and methods

Gravity core DA17-NG-ST07-73G (79.068° N 11.903° W; hereafter 73G) was collected on September 16, 2017, at 385 m water depth in an isolated bathymetrical depression on the NE Greenland shelf during the NorthGreen2017 Expedition (Fig. 1; Seidenkrantz et al., 2018). In addition, we collected Rumohr core DA17-NG-ST07-72R (79.072° N 11.888° W, 384.6 m water depth; hereon 72G) with an intact sediment-water interface from the same station, which provided information for the age model of the gravity core. Temperature and salinity of the water column were measured in situ by a Seabird CTD (Conductivity-Temperature-Depth) profiler (Seabird SBE911 plus system equipped with two temperature and conductivity sensors) before the deployment of the gravity corer, with the core site being identified based on Innomar® subbottom profiler data obtained prior to coring.

The 410 cm long gravity core (73G) and the 88 cm long Rumohr core (72R) were stored at ca 3 °C before they were split lengthwise. The archive halves were scanned with an Itrax X-ray fluorescence core scanner (Cox Analytical) at the Department of Geoscience, Aarhus University, Denmark. The scan was conducted with a Molybdenum tube (other settings: step size: 200 µm; exposure time: 10 s; voltage: 30 kV, current: 30 mA). The core scanning provided a line scan image, a 2-cm-wide radiographic image of the centre of the core (voltage: 60 kV; current: 45 mA; exposure time: 2000 ms) and high-resolution analysis of bulk geochemical composition. From the results of the X-ray fluorescence scanning, we show the elements Si, K, Ti and the Ca/Fe ratio on Fig. 3.

The age-depth model of core 73G is based on six Accelerator Mass Spectrometry radiocarbon (AMS $^{14}$C) dates, of which five are of mixed benthic foraminifera and one is of a worm tube lining. AMS $^{14}$C dating was performed at the Laboratory of Ion Beam Physics, ETH Zürich, Switzerland. Radiocarbon ages were calibrated using the Marine20 calibration dataset (Heaton et al., 2020) and a local marine reservoir age correction ($\Delta R$) of $0 \pm 50$ years (Reimer and Reimer, 2001). We are aware that Heaton et al. (2020) state that the Marine20 is not suitable for polar regions. However, the same was true, just not explicitly stated, for the earlier marine radiocarbon age calibration curve Marine13 (Reimer et al., 2013), which has been commonly used in Arctic palaeoceanography (e.g., Perner et al., 2015; Seidenkrantz et al., 2019; Syring et al., 2020a). The presence of sea ice in polar regions impacts the local reservoir age, and therefore there is added uncertainty; this issue is not resolved by using an older calibration dataset. Since its publication, the Marine20 has been widely used in the Arctic realm (e.g., Farmer et al., 2021; Altuna et al., 2021). Moreover, for this specific Holocene reconstruction, the differences between using Marine13 and Marine20 are much smaller than the associated uncertainties (see Table A1 in the appendices).
The age-depth model was made with the OxCal v4.4 software (Ramsey, 2008), using a P-sequence depositional model. Beyond the lowest radiocarbon date, the model is extrapolated toward the bottom of the core. The age of the upper part of the core is constrained by correlation to Rumohr core 72R by selected XRF elements (appendices, Fig. A1), to determine the amount of top sediment loss during gravity coring.

Samples for foraminiferal, ice-rafted debris (IRD) and stable isotope analyses were taken at 5 cm intervals at the Department of Geoscience, Aarhus University. The samples were weighed, wet-sieved with distilled water through sieves of 1000, 100 and 63 µm, dried at 40 °C for 24 hours and weighed again. For the foraminiferal assemblage analyses both planktic and benthic (calcareous and agglutinated) species were counted and identified to species level under a stereomicroscope for the size fractions of 63-100 µm and >100 µm; counts were subsequently combined prior to percentage calculations. When the samples contained excessive material, prior to counting they were split into equal parts containing >300 specimens, using a microsplitter. In these cases, foraminiferal tests of one part were counted and identified, and the total number of foraminifera was calculated. A minimum of

300 benthic specimens were identified for each sample, except for four samples (10-11 cm, 15-16 cm, 30-31 cm, 195-196 cm). These four samples contained only between 242-296 specimens. Absolute foraminiferal concentrations were calculated as individuals/g wet sediment. Relative abundances of both benthic agglutinated and calcareous species were calculated as a percentage of the total benthic (calcareous and agglutinated) foraminiferal fauna. However, diagrams of the separate percentage calculations of the two benthic foraminiferal groups (agglutinated species of total agglutinated assemblage and calcareous species of total calcareous assemblage) are shown in the appendices (Figs. A2 and A3), as are diagrams of the same species as number of individuals/g wet sediment.

After counting, we placed selected benthic calcareous foraminifera species into groups according to their main environmental preferences; for details see the discussion (chapter 5.2) and Table A3 in the appendices. The Atlantic Water group includes *C. neoteretis* and *P. bulloides*. The chilled Atlantic Water group includes *I. norcrossi* and *M. barleeanum*. The Arctic Water group includes *S. horvathi* and *E. arctica*; these species are also found in connection to sea ice. The sea-ice edge group includes *S. feylingi*.

For stable isotope analysis 5-200 specimens of the benthic foraminiferal species *Elphidium clavatum* were picked from every sample. The number of picked tests was restricted by the number of available, clearly identifiable tests without any corrosion or non-typical shapes. The specimens belonged mainly to the 100-1000 µm fraction, supplemented with few specimens from the 63-100 µm fraction, when necessary. The oxygen and carbon isotope analysis of foraminiferal calcite was performed at the Leibniz Laboratory for Radiometric Dating and Stable Isotope Research at the Christian-Albrechts-University of Kiel using a MAT253 (Thermo Scientific) mass spectrometer system and a Kiel IV carbonate preparation device, with an analytical accuracy of <0.08 ‰ for $\delta^{18}O$ and <0.05 ‰ for $\delta^{13}C$. Results are expressed in δ notation referring to the PDB (Pee Dee Belmnite) standard, while using NBS-19 and IAEA-603 carbonate standards.

The content of IRD was calculated at 5 cm sample intervals using the dry weight of the size fraction >1000 µm divided by the wet weight of the bulk sample.

Grain size distribution was measured on 42 samples (taken at 10-cm intervals) using a laser diffraction particle sizer (Sympatec Helos) at the Department of Geoscience, Aarhus University, and grouped into three fractions: sand (>60 µm), silt (2-60 µm) and clay (<2 µm), according to the available particle size intervals.

## 4 Results

### 4.1 Core description

Core DA17-NG-ST07-73G consist entirely of olive grey (5Y 4/1) marine silt with some clay, except the lowermost part of the core. The sediment of the lowest 40 cm of the core is much coarser, containing up to 51 % sand; it is also darker in colour (Fig. 3). In the rest of the core the sand fraction is on average 1.5 % and the clay : silt : sand ratio does not show any significant changes.

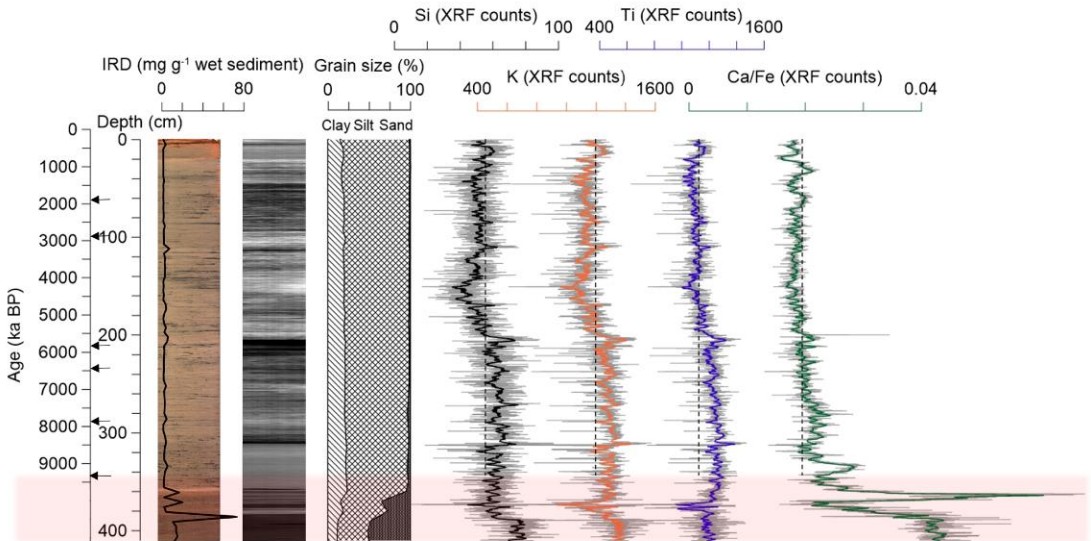

Figure 3: Results of the sedimentological and XRF core scanning analyses of core DA17-NG-ST07-73G. From left to right: photograph of the core with the amount of IRD (black line), radiograph, grain size analysis and XRF counts of selected elements. Black dashed lines in the XRF curves represent the mean of the values measured throughout the core. Black arrows next to the left y-axis (age) mark radiocarbon dates. The bottom 65 cm of the sediment core is marked with rose-pink colour. The photograph and radiograph of the core were horizontally stretched.

## 4.2 Chronology

The age model was developed using six AMS [14]C measurements on mixed benthic foraminifera and on an organic worm tube lining from core 73G (Table 1). In order to evaluate potential loss of core top sediment during the coring process, we compared the XRF data of gravity core 73G and Rumohr core 72R (Fig. A1 in appendices), which indicated that 12 cm sediment was missing from the top of core 73G; this information was used when constructing the age model of core 73G. This age model suggest that the 410 cm long sediment core covers the last ~11.2 ka BP (Fig. 4). However, the results of the grain size, IRD and X-ray fluorescence analysis indicate that beyond the lowest radiocarbon-dated sample (at 345 cm depth, 9.4 ka BP) the sediments change significantly (Fig. 4). In the bottom 65 cm of the core, preliminary analysis of foraminiferal content revealed the presence of species indicating a Pliocene or early Pleistocene age (Feyling-Hanssen, 1976; Feyling-Hanssen, 1980; Feyling-Hanssen et al., 1983), in addition to the normal in situ foraminiferal assemblage (see chapter 4.4.1). We suggest that these Plio-/Pleistocene foraminifera are reworked into a glaciomarine setting. Due to this very different environment and the lack of [14]C dates in this interval, the age model of the bottom 65 cm of core 73G is currently uncertain. Therefore, the main focus of this paper is on the last 9.4 ka.

Table 1: List of radiocarbon dates and modelled ages in core DA17-NG-ST07-73G. All dates were calibrated using the Marine20 calibration curve and ΔR=0 ± 50 years. Comparison between results from calibration with Marine20 vs Marine13 is shown in Table A1 in the appendices.

| Lab. ID | Depth (cm) | Material | Radiocarbon age (yr BP) | Error (yr) | Calibrated 2-sigma (BP) | | | Modelled 2-sigma (cal yr BP) | | |
|---|---|---|---|---|---|---|---|---|---|---|
| | | | | | from | to | median | from | to | median |

| | | | | | | | | | |
|---|---|---|---|---|---|---|---|---|---|
| ETH-95387 | 70.5 | Mixed benthic foraminifera | 2475 | 60 | 2200 | 1705 | 1952 | 2161 | 1817 | 1989 |
| ETH-95388 | 110.5 | Mixed benthic foraminifera | 3275 | 70 | 3190 | 2715 | 2935 | 3165 | 2783 | 2972 |
| ETH-110893 | 226 | Worm tube lining | 5645 | 30 | 6029 | 5622 | 5835 | 6031 | 5678 | 5857 |
| ETH-95389 | 250.5 | Mixed benthic foraminifera | 6015 | 70 | 6468 | 5994 | 6241 | 6643 | 6275 | 6458 |
| ETH-95390 | 300.5 | Mixed benthic foraminifera | 7595 | 70 | 8097 | 7654 | 7870 | 8151 | 7761 | 7942 |
| ETH-95391 | 345.5 | Mixed benthic foraminifera | 9015 | 70 | 9823 | 9295 | 9536 | 9602 | 9125 | 9383 |

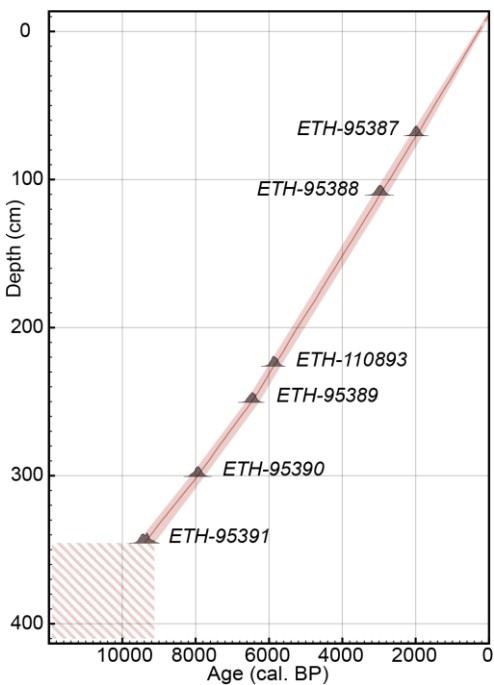

**Figure 4: Age-depth model of core DA17-NG-ST07-73G based on six AMS 14C dates and on the comparison with Rumohr core DA17-NG-ST7-72R. The shaded area around the line illustrates the 2-sigma uncertainty range of the model. The hatched box in the bottom of the core indicates an interval containing many reworked microfossils, which is therefore yet of uncertain age (see pink shading on Fig. 3).**

## 4.3 X-ray fluorescence

Throughout the whole core, Si, K and Ti follow the same pattern. After a drop in values in the lowermost section of yet uncertain age, all three records show relatively constant values until ~318 cm depth (ca. 8.5 ka BP). At 316 cm (ca. 8.4 ka BP) all three elements displays peak counts, followed by a decrease until 161 cm (ca. 4.2 ka BP). There is a short-term drop in all three element counts at 161 cm depth (ca. 4.2 ka BP), followed by a steady increase until 125 cm (ca. 3.2 ka BP) and relatively constant values until the top of the core (Fig. 3). The Ca/Fe ratio follows a steadily decreasing trend throughout the core, with slightly stronger fluctuations from 345 cm until ~270 cm (ca. 7 ka BP) and in the top 55 cm (last ~1.5 ka BP). The record shows a pronounced peak between 335 and 338 cm (ca. 9.1 ka BP), and two minor peaks at ~213 cm (ca. 5.5 ka BP) and at ~34 cm (ca. 0.9 ka BP) (Fig. 3).

**4.4 Foraminiferal analysis**

**4.4.1 410-345 cm core depth**

The bottom 65 cm of the core are characterised by a mixture of two assemblages. An assemblage of relatively small, well-preserved specimens of Arctic species include *Cassidulina reniforme, Elphidium clavatum, Islandiella norcrossi*, and *Melonis barleeanum.* This assemblage is mixed with large specimens of *Cibicidoides grossus, Cibicides scaldiciensis, Islandiella inflata, Cassidulina teretis*, and *Elphidium funderi*, which are extinct since the end Pliocene to Mid Pleistocene (Feyling-Hanssen, 1976; Feyling-Hanssen, 1980; Feyling-Hanssen et al., 1983; Seidenkrantz, 1995) and by species that are only present in temperate to tropical regions (*Bulimina marginata*, *Faujasina* sp.). The assemblage also contain some unusually large specimens of *E. clavatum, Haynesina orbiculare*, and *Islandiella islandica*.

**4.4.2 345-0 cm core depth**

We counted in total 70 samples, and identified 65 benthic (44 calcareous and 21 agglutinated) and three planktic foraminiferal species (Table A2 in the appendices) in the top 345 cm of core 73G. Both benthic and planktic specimens were well-preserved and showed minor or no signs of post-mortem dissolution of the tests.

The foraminiferal content is relatively low throughout the core, the benthic foraminiferal concentration varies between 17 and 158 individuals (ind.) g$^{-1}$ wet sediment (sed.) (calcareous 2-128 ind. g$^{-1}$ sed., agglutinated 10-55 ind./g sed.) (Fig. 5).

The concentration of planktic foraminifera is extremely low throughout the core, it lies between 0 and 13 ind. g$^{-1}$ sed. Planktic foraminifera are represented primarily by the polar species *Neogloboquadrina pachyderma*, occasionally accompanied in the interval between 340-130 cm (ca. 9.4-3.4 ka BP) by specimens of the subpolar species *Turboratalita quinqueloba* and *Neogloboquadrina incompta*. The overall occurrence of planktic foraminifera is averaging 2 % of the total foraminiferal fauna.

On average, benthic agglutinated foraminifera account for 65 % of the total foraminiferal assemblage. From the end of the record until 210 cm (ca. 5.4 ka BP) the calcareous species periodically outnumber the agglutinated ones. However, from 210 cm until the top of the record, agglutinated foraminifera continuously dominate the benthic assemblage (Fig. 5). Throughout the core, the most abundant benthic agglutinated species are *Portatrochammina bipolaris*, followed by *Ammoglobogerina globigeriniformis*, representing on average 42 % and 16 % (respectively) of the benthic agglutinated assemblage, and 27.5 % and 10.5 % of the total benthic assemblage. They are both continuously present throughout the core and their relative abundances do not show strong fluctuations (Fig. 6).

Benthic calcareous foraminifera represent on average 33 % of the total foraminiferal assemblage. The most abundant benthic calcareous species are *C. reniforme*, *Cassidulina neoteretis, E. clavatum* and *Stetsonia horvathi* (representing on average 8.7%, 6.4%, 5.9% and 4.4% of the whole benthic assemblage, respectively). From these four species the abundances of *E. clavatum* shows the least variability throughout the record (Fig. 5).

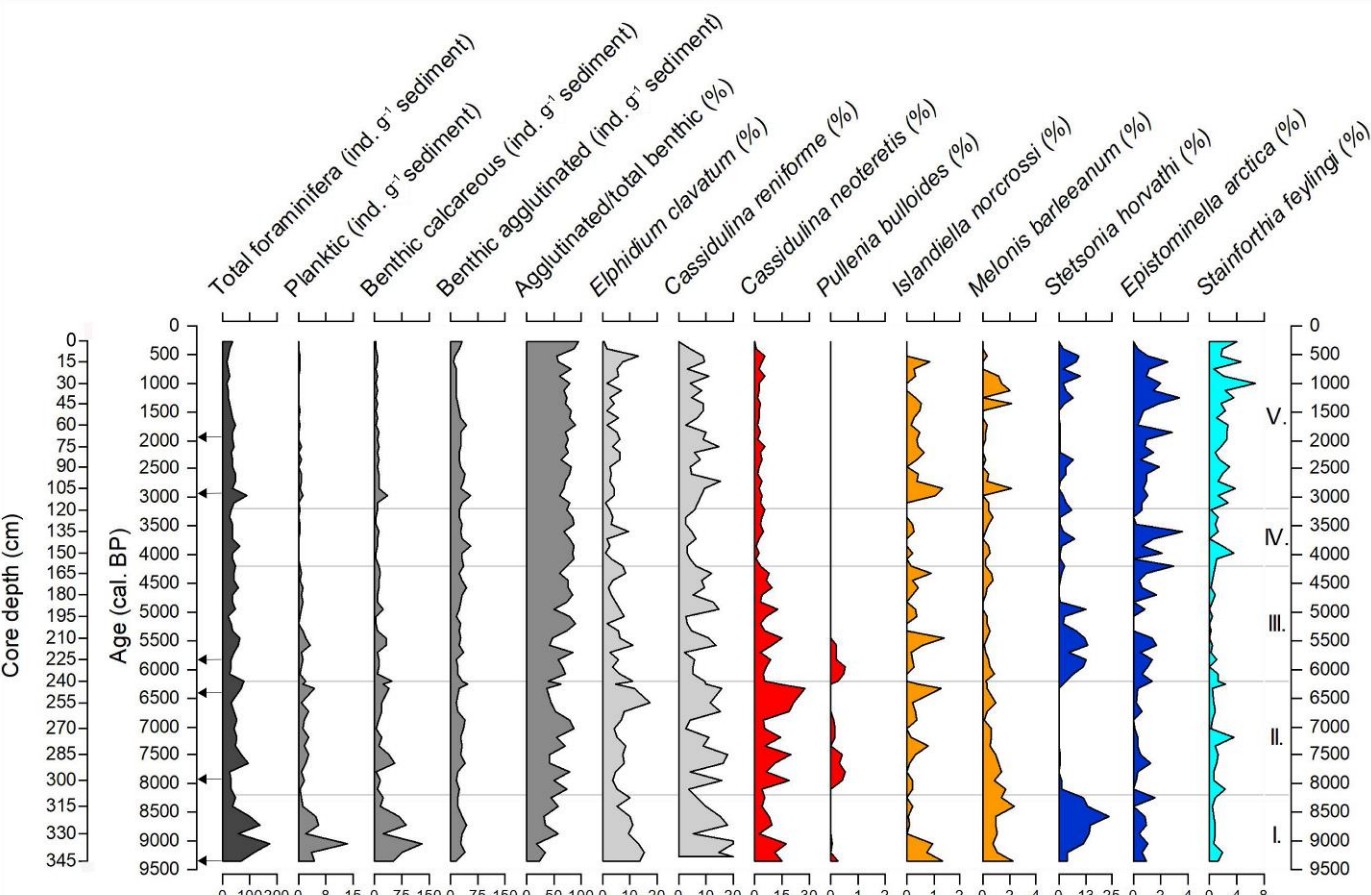

**Figure 5: Foraminiferal concentrations and relative abundances of nine selected calcareous benthic species (expressed as a percentage of total benthic foraminiferal content) versus calibrated age along sediment core DA17-NG-ST7-73G. The depicted species were chosen in order to show changes in the water masses. Red colour represents species that indicate warm Atlantic Water inflow; orange represents species that indicate in the Arctic recirculated, chilled Atlantic Water influence; dark blue represents Arctic Water species often living beneath perennial or near-perennial sea ice; light blue represents a species that indicates sea ice edge conditions. Ecozones (I-V) are shown on the right side of the figure. Black arrows next to the left primary y-axis (Age) mark radiocarbon dates. Note that the x-axes have different scaling.**

In the following sections, we describe foraminiferal assemblage zones (ecozones) in the dated section of the core (345-0 cm core depth). These zones were defined by visual evaluation and interpretation of the species abundances. Here it should be mentioned that also species with relatively low abundances can be significant for the environmental interpretations. Boundaries were placed where major changes occurred in the relative abundances of the most important benthic calcareous and agglutinated species, indicating changes in the environment (Figs. 5 and 6). Unless otherwise specified, all relative frequencies are provided as average values for the interval.

### 4.4.3 Ecozone I (345-310 cm; ca. 9.4-8.2 ka BP)

This ecozone is characterised by the highest content of total foraminifera (171 ind. g$^{-1}$ sed.), including the highest concentrations of planktic (13 ind. g$^{-1}$ sed.) and benthic calcareous (128 ind. g$^{-1}$ sed.) foraminifera throughout the core. These three categories show two peaks in this interval: the first and bigger at ca. 9 ka BP and the second, smaller one at ca. 8.7 ka BP. We find in this interval the lowest agglutinated/calcareous ratio (unshown), as the calcareous specimens almost continuously outnumber the agglutinated ones. The average concentration of agglutinated foraminifera is 29 ind. g$^{-1}$ sed. in this zone, and it does not vary significantly throughout the other intervals either (27-29 ind. g$^{-1}$ sed.). The benthic calcareous assemblage is dominated by *C.*

*reniforme* (relative abundance on average 14 %), followed by *S. horvathi* (12 %), *E. clavatum* (11 %) and *C. neoteretis* (9 %), as important accessory species.

### 4.4.4 Ecozone II (310-245 cm; ca. 8.2-6.2 ka BP)

In this ecozone, we recognise a distinctive decrease in planktic and benthic calcareous foraminiferal concentrations, and as a consequence, a higher benthic agglutinated/calcareous species ratio. The relative abundance of *C. neoteretis* promptly increases, and it becomes the dominating species of the benthic calcareous fauna in this interval (highest relative abundance 27 %, on average 13 %), followed by *C. reniforme* (10 %). *Pullenia bulloides* suddenly appears in the record becoming an important accessory species, and, at the same time, the relative abundance *S. horvathi* drastically decreases and it remains around 1 % during the whole period. Even though the relative abundances of *E. clavatum* decrease compared to the previous interval, this species shows its highest relative abundance throughout the core in this ecozone at 255 cm (17 %; ca. 6.5 ka BP).The relative abundance of agglutinated species *Adercotryma glomerata* increases from 310 cm on and remains on the same level throughout this zone.

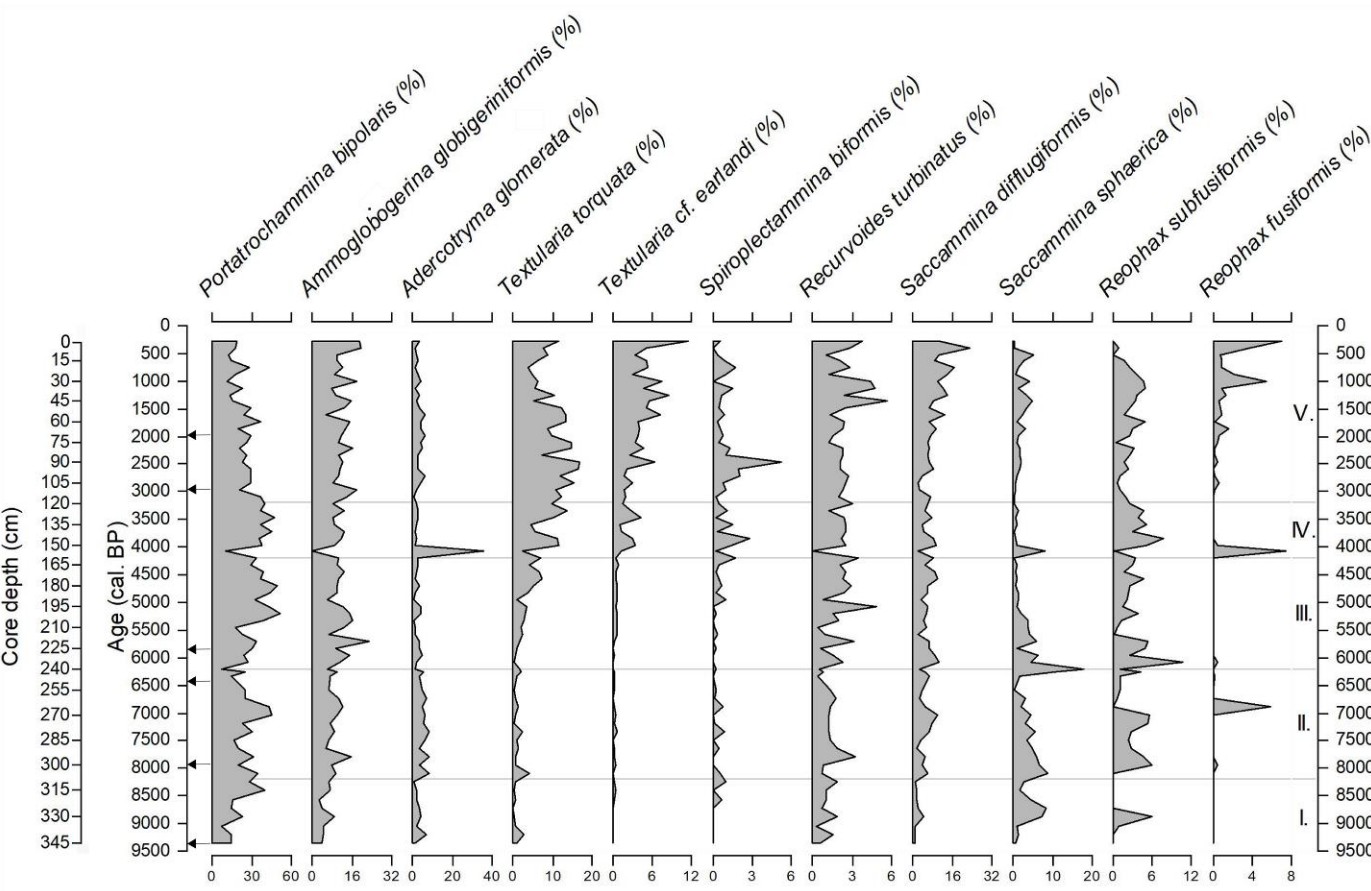

**Figure 6: Relative abundances of the most abundant (>5% in at least one sample) agglutinated benthic foraminiferal species (expressed as a percentage of total benthic foraminiferal content) versus calibrated age along sediment core DA17-NG-ST07-73G. Ecozones (I-V) are shown on the right side of the figure. Black arrows next to the left primary y-axis (Age) mark radiocarbon dates. Note that the x-axes have different scaling.**

### 4.4.5 Ecozone III (245-165 cm; ca. 6.2-4.2 ka BP)

The concentrations of planktic and benthic calcareous foraminifera continue to decrease in this ecozone. From 245 cm, the relative abundance of *C. neoteretis* drops back to an average of 7 %, while the relative abundance of *S. horvathi* rises (from 1 % to 7 %). Although still only found in low numbers, *Epistominella arctica* also increases (from 0.4 % (previous interval) to 0.8 %), just like *Textularia torquata*, while the relative abundances of *A. glomerata* decreases.

### 4.4.6 Ecozone IV (165-125 cm; ca. 4.2-3.2 ka BP)

The base of this ecozone is defined by the drastic decrease in benthic calcareous foraminiferal concentrations (from 15.7 to 8.4 ind. g$^{-1}$ sed.), and increased relative abundances of *E. arctica* and *Stainforthia feylingi*. *C. neoteretis* drops dramatically (to 3 %), just like *C. reniforme* that reaches its lowest relative abundance in this zone (5 %). On the other hand, agglutinated species *A. glomerata* and *Reophax fusiformis* shows a drastic peak at the beginning of this ecozone, and the relative abundances of *T. torquata*, *Textularia* cf. *earlandi* and *Spiroplectammina biformis* starts to increase as compared to the previous interval.

### 4.4.7 Ecozone V (125-0 cm; ca. 3.2-0.3 ka BP)

The concentrations of planktic foraminifera and the relative abundances of *C. neoteretis* further decrease and reach their lowest level throughout the core, while *C. reniforme* shows an increase compared to ecozone IV, just like *S. feylingi*. The relative abundances of agglutinated species *T. torquata*, *T.* cf. *earlandi* and *S. biformis* continue to increase, while *Saccamina difflugiformis* shows a steep rise unique to this interval.

### 4.5 Stable isotopes

From the 70 analysed samples, the results of three samples were not accepted: one (180 cm) due to technical problems, and two gave too low signals due to small sample size (0 cm and 130 cm). The sample at 140 cm gave a comparatively very low δ$^{18}$O (2.25 ‰) and a very high δ$^{13}$C value (-1.07 ‰), but these outliers were nonetheless accepted.

Except for the one outlier, the variation of δ$^{18}$O values is rather small throughout the core (between 2.85 and 3.69 ‰, with a mean of 3.29 ‰), and the values remain relatively constant, albeit with a slight increase towards the top of the core. The δ$^{13}$C values show a constant increase from the bottom to the top of the core, varying between -1.07 and -2.61 ‰ (Fig. 7).

## 5 Discussion

### 5.1 Origin of Atlantic-sourced water at the core site

Although the study site is located on the central flow route of the EGC, the path of the influx of the Atlantic-sourced subsurface water, which is clearly identified as the bottom layer in the CTD profiles (Fig. 2), is less clear. The CTD profile with bottom water temperatures below 1 °C and salinities of 34.8 suggest that the Atlantic-sourced subsurface water is closer in character to the water found in the Westwind Trough than that of the Atlantic-sourced water presently seen in the Norske Trough (c.f., Budéus et al., 1997). This may indicate that cool Arctic Atlantic Water is the primary source of the Atlantic subsurface waters, rather than the warmer Return Atlantic Water (cf. Rudels et al., 2005; Schaffer et al., 2017). The Atlantic-sourced water may thus reach our study site via the Westwind Trough, but may also be present due to wind-driven upwelling. This process leads to considerable modification of local water masses and shelf break exchanges (e.g., Estrade et al., 2008; Kirillov et al., 2016).

### 5.2 Environmental significance of foraminiferal assemblages

In order to be able to describe the changes in water masses over time on the NE Greenland shelf, we place selected benthic calcareous foraminiferal species into groups that are based on environmental preferences of the species (Table A3 in appendices). The Atlantic Water group includes *C. neoteretis* and *P. bulloides*. These species indicate warm and saline AW inflow underneath cold and low salinity surface waters (e.g., Mackensen and Hald, 1988; Seidenkrantz 1995; Rytter et al., 2002; Jennings et al., 2004; Jennings et al., 2011; Cage et al., 2021). The chilled Atlantic Water group includes *I. norcrossi* and *M. barleeanum*. These species have been previously linked to cool AW (e.g., Slubowska-Woldengen et al., 2007; Perner et al., 2011; Perner et al., 2015; Cage et

al., 2021), and we use them to represent the relative contribution of chilled AW recirculated in the Arctic Ocean to the EGC. The Arctic Water group includes *S. horvathi* and *E. arctica*. Both species at present live beneath perennial to near-perennial sea ice, in the deeper part of the Arctic Ocean, thus indicating cold, Arctic originated deep-waters (e.g., Green, 1960; Lagoe, 1979; Wollenburg and Mackensen, 1998; Jennings et al, 2020), although while *S. horvathi* is found beneath true perennial sea ice, *E. arctica* seems to thrive in high-productivity areas in open-water holes in the ice or at the sea-ice edge (Wollenburg and Mackensen, 1998; Jennings et al, 2020). *Cassiduina reniforme* is also an Arctic species, although it requires stable conditions (e.g., Hald and Korsun, 1997; Consolaro et al., 2018), and in some studies it has been considered linked to highly chilled Atlantic Water (Ślubowska et al., 2005; Hansen et al., 2020). *S. feylingi* is a true sea-ice edge indicator species that tolerates unstable conditions (Knudsen and Seidenkrantz, 1994; Seidenkrantz, 2013); its increase may refer to the location of a sea-ice margin at the study site. Co-occurrence of *S. feylingi* and *E. arctica* thus suggests the presence of a nearby sea-ice edge in an otherwise near-perennial sea-ice region. Moreover, we use in our interpretation the abundances of the agglutinated species *A. glomerata*, *T. earlandi*, *T. torquata*, *S. biformis* and *S. difflugiformis*. *A. glomerata* is often correlated with Atlantic sourced waters in high latitudes (e.g., Hald and Korsun, 1997, Lloyd 2006; Perner et al., 2012). Many studies find *T. earlandi*, *T. torquata* and *S. biformis* in areas influenced by cold, low-salinity Arctic water, often in glaciomarine environments (e.g., Jennings and Helgadottir, 1994; Korsun and Hald, 2000; Perner et al, 2012; Perner et al., 2015, Wangner et al., 2018). *S. difflugiformis* is described as closely associated with Polar Water on the East Greenland shelf and in fjords (Jennings and Helgadottir, 1994).

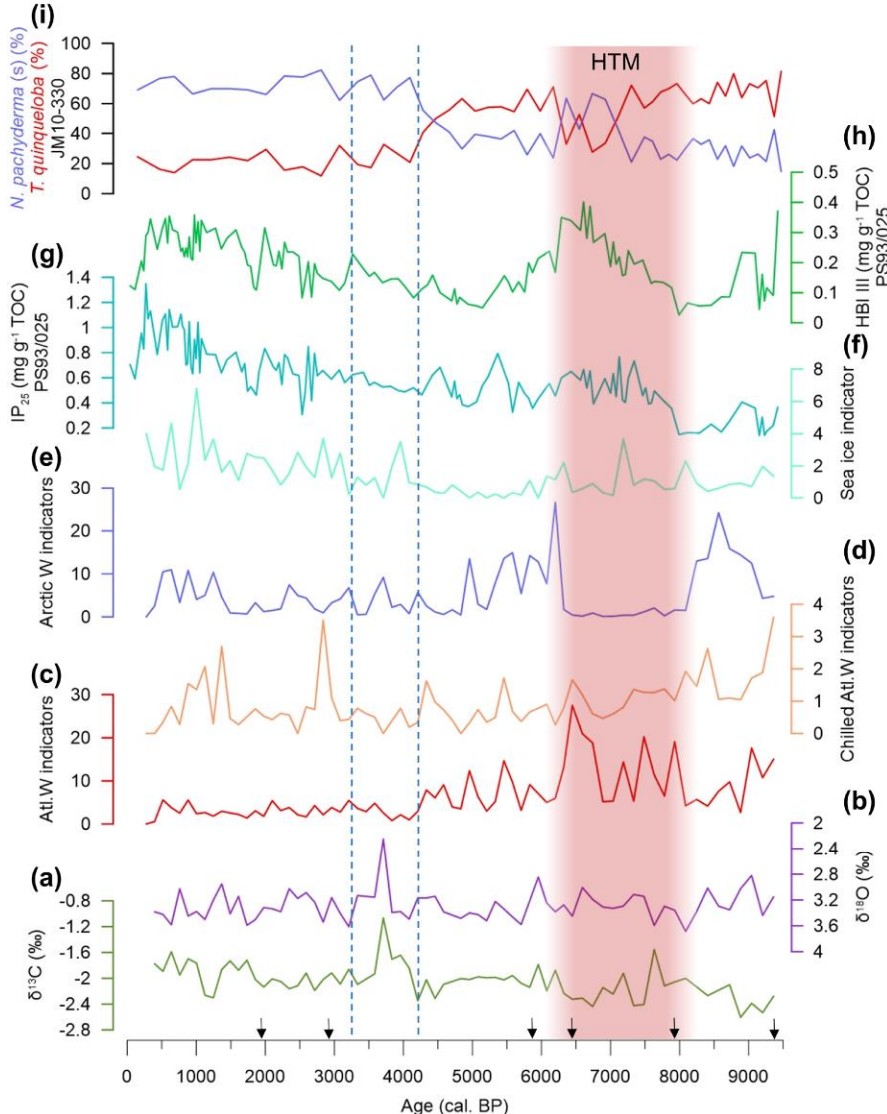

**Figure 7: From the bottom to the top: δ¹³C (a) and δ¹⁸O (b) results and relative abundances of the four calcareous benthic foraminifera groupings (c-f) (based on environmental preferences of the contained species, see Fig. 5 and Appendix Table A3) from this study (core 73G), compared to other proxy records from the NE Greenland shelf and western Svalbard. IP₂₅ (g) and HBI III (h) from core PS93/025 (Syring et al., 2020a). Relative abundances of planktic polar species *Neogloboquadrina pachyderma* (sin.) and subpolar species *Turborotalita quinqueloba* (i) from core JM10-330 (Consolaro et al., 2018). HTM: Holocene Thermal Maximum. Black arrows on the x-axis mark radiocarbon dates.**

## 5.3 Palaeoenvironmental interpretation

### 5.3.1 Bottom 65 cm of the sediment core

The combined presence of an assemblage of well-preserved small tests of Arctic foraminiferal species and many extinct Pliocene to early-Pleistocene specimens, today found in Plio-Pleistocene assemblages from Baffin Island and Peary Land (Feyling-Hanssen, 1976; Feyling-Hanssen, 1980; Feyling-Hanssen et al., 1983; Funder et al., 2001), suggest that these Plio-Pleistocene foraminifera have been reworked from older deposits into a glaciomarine environment with *in situ* Arctic foraminifera. The closest known Plio-Pleistocene deposit is the Kap København Formation from Peary Land, northern Greenland (Funder et al., 2001; see Fig. 1b), but

380 it is unknown if any such deposits may be found outcropping to the sea floor off NE Greenland. Reworking of older sediments during glacier ice retreat has previously been observed (e.g., Seidenkrantz et al., 2019). Therefore, we here suggest that the breakup and significant retreat of a nearby glacier caused reworking of older sediments and thus, the lowermost section of sediment from before 9.4 ka was deposited during a period of glacier retreat. However, further studies are required to resolve this issue.

**5.3.2 Early Holocene (ca. 9.4-8.2 ka BP; ecozone I)**

385 This interval was characterised by high percentages of *S. horvathi*, which has previously been considered linked to high sea-ice concentrations (e.g., Green, 1960; Lagoe, 1979; Wollenburg and Mackensen, 1988; Jennings et al., 2020) combined with the presence of Atlantic Water indicator species such as *C. neoteretis, I. norcrossi* and *M. barleeanum*. This combination of Atlantic-sourced bottom water and sea ice suggest a highly stratified water column in the Early Holocene (Fig. 5). Cold, heavily sea ice-loaded surface waters entrained in PW of the upper EGC characterised the upper water column, while at the same time the influx

390 of warm Atlantic-sourced water from either the RAC or the AAW (or a combination of both) was also strong. When compared to the data from the top of the core, the water column stratification between the cold and warmer waters was likely more pronounced than today. Relatively high counts of the terrestrially-derived elements Si, K and Ti, (Ren et al., 2009) together with relatively low $\delta^{18}O$ values may indicate increased meltwater influence from the Greenland Ice Sheet (Figs. 3 and 7) that may have reached the bottom through melt-water plumes. Alternatively, the low $\delta^{18}O$ values seen here may be a consequence of the warmer bottom

395 waters.

The interval was also likely characterised by high surface water productivity, as shown by high concentrations of planktic and benthic calcareous foraminifera and the presence of *E. arctica*, which thrives in high-productivity environments (Wollenburg and Kuhnt, 2000). It is also supported by a pronounced peak in Ca/Fe ratio, which is considered as a general proxy for palaeo-productivity (e.g., Vare et al., 2009) and low and slowly rising benthic $\delta^{13}C$ values that can also be related to high level of

400 bioproductivity, due to decomposition of isotopically light organic carbon in the lower part of the water column. High surface water productivity may sound contradictory to extensive sea-ice cover, however, the planktic foraminifera may have been transported to the site in the subsurface water layer from a nearby more open-ocean site, suggesting increased surface water-bioproductivity in the general region. We propose that high productivity combined with sea ice indicates that our site was located west of the Polar Front, but that the Polar Front was situated in close proximity to our location in this time interval, which would

405 have facilitated a high productivity. It should be noted that in the XRF data, we recognise a short-lived, but clear peak in terrestrially-derived sediments around 8.5-8.4 ka BP (Fig. 3). This may be potentially linked to increased iceberg release during the so-called 8.2 ka event (i.a., Alley et al., 1997; Barber et al., 1999), however, this signal seems to be only reflected in the results of the sedimentological analyses. It is not recognizable in the foraminiferal assemblage changes/stable isotope results, although the lower temporal resolution of the latter may have prevented us from identifying any changes.

410 Our suggestions of cold water and sea ice at the surface combined with Atlantic-sourced bottom waters is in accordance with a recent reconstructions from the NE Greenland shelf (Syring et al., 2020a; Zehnich et al., 2020; core PS93/025). The study of Syring et al. (2020a) reveals a cold interval with increased sea-ice cover between 9.3 and 7.9 ka BP, shown by lower concentrations of marginal sea-ice proxy HBI III and IP$_{25}$, interpreted as low sea-ice algal productivity caused by an expansion of the sea-ice cover (Fig. 7). The stable isotope results of Zehnich et al. (2020) point to a relatively thick subsurface Atlantic water mass. Foraminiferal

415 records from the Southeast Greenland shelf (Jennings et al., 2011; cores MD99-2322 and MD992317) suggests cold and unstable sea-surface conditions between 9.4 and 8.1 ka BP. Moreover, cool deep-water conditions in the eastern Fram Strait during this period (Consolaro et al., 2018; core JM10-330) point to a weakened RAC, suggesting that the strong water column stratification on the NE Greenland shelf was due to increased AAW influx.

The cold, heavily sea ice-loaded surface waters and strongly stratified water column that characterised our study site on the NE Greenland shelf between 9.4 and 8.2 ka BP can be explained as a consequence of the extensive melting of the Greenland Ice Sheet. Early Holocene warming of the circum-Arctic region (Kaufmann et al., 2004) led to enhanced Greenland Ice Sheet and glacier melting around the Arctic (e.g., Solomina et al., 2015). During the Last Glacial Maximum (~26-19 ka BP) the Greenland Ice Sheet extended onto the shelf offshore NE Greenland, perhaps even reaching the shelf break (Bennike and Weidick, 2001; Arndt et al.,

2017). However, as a consequence of rising air temperatures due to the orbitally-forced Northern Hemisphere summer insolation maximum (reported at approximately 10 ka BP; Andersen et al., 2004; Jansen et al., 2008), the deglaciation of the outer coast and retreat of the ice margin to its present location occurred already between 11.7 and 9.3 ka BP (Larsen et al., 2018); the ice shelf of 79NG rapidly retreated through the fjord between 9.6 and 7.5 ka BP (Syring et al., 2020b). The drastic ice recession of the early Holocene produced an extended meltwater surface layer in the Greenland region prior to 8.6 ka BP (Seidenkrantz et al., 2013).

The extensive melting of the Greenland Ice Sheet was strong enough to act as negative feedback to the Early Holocene warming, and delayed the HTM with 2 kyr at our location compared to the eastern parts of the Nordic Seas (Blaschek and Renssen, 2013). The strong freshwater discharge might have also weakened the recording of the 8.2 ka cooling event at our location, although a short-term increase in terrestrial deposits suggests transport of sediments from land via meltwater or icebergs during this event. This 8.2 ka cold event has been described in a number of palaeoceanographic archives from Greenland and the northern North

Atlantic (e.g., Risebrobakken et al., 2003; Ellison et al., 2006; Rasmussen et al., 2007), and it is believed to be connected to the collapse of the Laurentide Ice Sheet and drainage of Lake Agassiz (e.g., Alley et al., 1997; Barber et al., 1999; Hillaire-Marcel et al., 2007; Hoffman et al., 2012). However, in many other records from areas that are mainly influenced by the EGC, WGC or the Baffin-Labrador Current system (Keigwin et al, 2005; Sachs 2007; Seidenkrantz et al., 2013), the 8.2 ka cooling event cannot be recognised; these sites were all characterised by the permanent presence of a low-salinity, cold surface layer at that time, which

may also explain the weak signal of this event at our site.

**5.3.3 Holocene Thermal Maximum (ca. 8.2-6.2 ka BP; ecozone II)**

The interval from 8.2 to 6.2 ka BP was characterised by the warmest bottom-water conditions of the Holocene on the NE Greenland shelf, as well as reduced sea-ice cover, as indicated by the almost complete absence of sea-ice indicator species and maximum abundances of Atlantic Water species and other species indicating stable bottom waters. *Cassidulina neoteretis* and *A. glomerata*

had their highest relative abundances during this interval (except for an extreme peak of *A. glomerata* at ~4 ka BP), and *P. bulloides* appeared around 8 ka BP in the record after a long absence (Figs. 5 and 6), suggesting highly stable bottom waters (Rytter et al., 2002). A concurrent significant decrease in planktic and benthic calcareous foraminiferal concentrations indicates a transition towards lower surface and subsurface water productivity, also shown in decreasing Ca/Fe element ratios. We suggest that this may be explained by a strengthening of the RAC, transporting RAW to our site and causing higher-than-present entrainment of warm

Atlantic-sourced water into the EGC. This is supported by several studies from the Fram Strait (Slubowska-Woldengen et al., 2007; core JM02-440; Werner et al., 2013; core MSM5/5-712; Werner et al., 2016; core MSM5/5-723; Consolaro et al., 2018; core JM10-330, see Fig. 1) that recorded strong AW flow and warm sea-surface conditions until 7-6.8 ka BP (Fig. 7). This strengthening would have also resulted in a stronger RAC. Moreover, in line with our findings, Müller et al. (2012) (core PS2641-4) and Jennings et al. (2011) (cores MD99-2322 and MD99-2317) reconstructed relatively warm conditions during the first part of the mid

Holocene on the middle East and Southeast Greenland shelf in surface and subsurface waters, respectively. Also further south in the Nordic Seas, thermal optimum-like conditions with warm sea surface temperatures prevailed until ca. 6 ka BP (Bauch et al., 2001; core PS1243).

The warmer sea-surface waters (as described in Müller at al., 2012 and Jennings et al., 2011), combined with the fact that our data indicate loss of sea ice and strengthened influx of Atlantic-sourced waters in the study area suggests that the Polar Front moved from our location to further north or further inland during this period. This time interval, which represents the warmest of our record, we here refer to as the Holocene Thermal Maximum (HTM).

Due to the warming during the HTM, the floating ice margin of 79NG decreased (Bennike and Weidick, 2001), but on the other hand, the fresh, less dense basal melt water of the glacier isolated the landfast sea-ice cover from the warm subsurface waters, and thus, stabilised the Norske Øer Ice Barrier (Mayer et al., 2000; Syring et al., 2020b). Parallel to the warming of the EGC (Jennings et al., 2011; Müller et al., 2012), the strengthening of the RAC seen at our site could be potentially linked to strengthening of the northern flow of AW, as recorded further south. Here the Irminger Current increased its strength, as recorded on the North Icelandic Shelf (Ran et al., 2006; Cabedo-Sanz et al., 2016).

### 5.3.4 Cooling after the HTM (ca. 6.2-4.2 ka BP; ecozone III)

Between 6.2-4.2 ka BP benthic foraminiferal assemblages again start to resemble those of the period prior to 8.2 ka BP, with increased percentage of *S. horvathi* and a decrease in the Atlantic water indicator *C. neoteretis* (Fig. 5). This suggests return to a more stratified water column with sea ice-loaded surface water and Atlantic-sourced subsurface waters. However, in contrast to Ecozone I, neither the benthic and planktic foraminiferal concentrations nor stable isotope data suggest increased bioproductivity. In line with the cooling of the bottom waters inferred from our data, the inflow of subsurface AW along West Spitsbergen, and thus the RAC, weakened after 7 ka BP (Consolaro et al., 2018; core JM10-330;), and a gradual cooling of surface waters started around 7-6.6 ka BP (Müller et al., 2012; core MSM5/5-712), accompanied by increased sea-ice cover (Slubowska-Woldengen et al., 2007; core JM02-440). Similar to our record, a sediment core retrieved north from our location (Zehnich et al., 2020; Syring et al., 2020a; core PS93/025) shows stronger vertical stratification after 7.2 ka BP (derived from benthic-planktic $\delta^{18}$O difference). After 5.5 ka BP this site experienced seasonally more extended sea-ice cover (derived from decreasing HBI III and increasing $IP_{25}$ concentrations; Fig. 7) and decreasing primary production (Zehnich et al., 2020; Syring et al., 2020a). Accordingly, at the same time, Perner et al. (2015) (core PS641-4) recorded the southward relocation of Polar Front to close to their site in the middle part of the East Greenland shelf. Giraudeau et al. (2004) (core MD99-2269) and Ran et al. (2006) (core MD99-2275) recorded increased influence of cool, low salinity polar waters of the EGC on the North Iceland Shelf between 6.5-3.5 ka BP and 6.8-5.6 ka BP, respectively. In the Iceland Basin a long-term cooling trend started at between 6.8 and 6.1 ka BP (Orme et al., 2018; Van Nieuwenhove et al., 2018; core DA12-11/2).

The cooling after the HTM started on the NE Greenland shelf with decreased, but still persistent inflow of subsurface AW via the RAC. The EGC became stronger, with sea ice-loaded surface waters and still relatively warm Atlantic-sourced subsurface waters (e.g., this study; Andrews et al., 1997; Jennings et al., 2002; Zehnich et al., 2020). Coincident with the expansion of the EGC, several studies from the Nordic Seas (e.g., Bauch et al., 2001; Hall et al., 2004; Hald et al., 2007) infer a weakening of the AMOC, increased water column stratification and less ventilated subsurface during this period. In line with a decreased flux of recirculating AW onto the NE Greenland shelf, the Northeast Greenland Ice Margin started to advance from its Mid Holocene minimum around 6 ka BP (Larsen et al., 2018). Reduced amounts of warm AW on the inner continental shelf most probably reduced the basal melting within the 79NG fjord and may have contributed to the re-advance of the ice shelf of 79NG seen from around 4.5 ka BP on (Bennike and Weidick, 2001; Syring et al., 2020b).

**5.3.5 Further cooling at the onset of the Late Holocene (ca. 4.2-3.2 ka BP; ecozone IV)**

Our record shows clear evidence of significant changes occurring at 4.2 ka BP. The period starts at our location with a sudden rise in the relative abundances of calcareous sea-ice indicator species *S. feylingi* and in the relative abundances of agglutinated species *T. earlandi*, *T. torquata* and *S. biformis* (Figs. 5 and 6), which are often connected to cold, low-salinity Polar Water (Jennings and Helgadottir, 1994; Korsun and Hald, 2000; Perner et al., 2012; Perner et al., 2015; Wangner et al., 2018). Furthermore, the interval is marked by lowest abundances of warm Atlantic Water species and very low concentrations (<1 ind./g) of planktic foraminifera. The benthic agglutinated/calcareous foraminiferal ratio increases and the terrestrially derived elements Si, K and Ti (Ren et al., 2009) show minimal values (Fig. 3), indicating very low meltwater influence in the EGC. Since the foraminiferal assemblage points to low bottom water temperatures, we interpret the slightly higher benthic $\delta^{18}O$ values and increasing $\delta^{13}C$ values after 4.2 ka BP (Fig. 7) as indicative of general cooling and weak influence of AW. Accordingly, the light $\delta^{18}O$ spike found in the middle of this interval cannot be attributed to sudden bottom water warming on the basis of the foraminiferal fauna. It is more readily explained by local brine rejection during sea ice formation, which can carry the light isotopic signal from the surface to the ocean bottom (e.g., Mackensen and Schmiedl, 2016). It should here be noted that apart from a single sample with increased relative frequencies of the warmer-water agglutinated species *A. glomerata* (Figs. 6 and A3), there is no evidence of a shorter-lived climate excursion that might be linked to the so-called 4.2 ka BP event (Weiss, 2017), only a general change in environment at this time. We suggest that increased PW at the surface of the EGC and reduction of warmer waters from the RAC at subsurface levels led to freshening and reduced stratification of the water column at our site; it may also have experienced (near) perennial sea-ice cover. The reduced strength of the RAC was likely caused by an overall decrease in AW transport of the North Atlantic and West Spitsbergen currents to the eastern Fram Strait, accompanied by surface-water cooling and increased sea-ice coverage after 5.5 ka BP (Hald et al., 2007; core MSM5/5-712; Werner et al., 2013; core PS1878; Telesinski et al., 2014b; core PS1878; Consolaro et al., 2018; core JM10-330; Fig. 7). The $\delta^{18}O$ results of Zehnich et al. (2020; core PS93/025) point to colder conditions and weak AW advection after 5 ka BP north from our location. Harsh conditions and strengthened EGC with permanent sea-ice cover dominated the central Greenland shelf as well (Perner et al., 2015; Kolling et al., 2017; core PS641-4). Further south on the shelf, Jennings et al. (2002) and Perner et al. (2016) (core JM96-1206) described increased PW influence in the EGC, glacier advance and iceberg rafting during this period.

**5.3.6 Neoglaciation (ca. 3.2-0.3 ka BP; ecozone V)**

The period from ca. 3.2 to 0.3 ka BP is characterised by a further increase in the relative abundances of sea-ice edge indicator species *S. feylingi* and the agglutinated species *S. difflugiformis*. Atlantic Water indicator *C. neoteretis* is only found in relatively low numbers, although there is some increase in species linked to chilled Atlantic Water (Figs. 5, 6 and 7). Continuously increasing benthic $\delta^{13}C$ values indicate strong ventilation of the water column (Fig. 7). The results point to cold and unstable conditions with minimum surface water productivity and increasing sea-ice cover at our location. During this period, the EGC was likely strengthened compared to previous times, with a thick layer of cold and fresh PW on the surface and recirculated AAW inflow from the Arctic at subsurface levels, as seen in the chilled Atlantic Water group.

Previous studies have indicated that during the late Holocene, the West Spitsbergen Current transported less and less heat (Slubowska-Woldengen et al., 2007; core JM02-440), causing less warm water to reach the EGC through the RAC. From West Spitsbergen (Müller et al., 2012; core MSM5/5-712; Consolaro et al., 2018; core JM10-330) and from several other locations in the Greenland Sea (e.g., Telesinski et al., 2014b, core PS1878) and West Greenland (e.g., Seidenkrantz et al., 2007; core 248260; Seidenkrantz et al., 2008; cores DA00-02P and DA00-03P) a general cooling trend has been reported, with more severe sea-ice

conditions. On the Iceland shelf strongly reduced Irminger surface water and low coccolith carbonate sedimentation indicate
extreme advection of polar waters and extended sea-ice development (Giraudeau et al., 2004; core MD99-2269).

The Neoglacial cold interval on the East Greenland shelf started with increased freshwater forcing from the Arctic Ocean (e.g.,
this study; Perner et al., 2015) and advance of the Greenland Ice Sheet (Andersen et al., 2004). According to model simulations of
Renssen et al. (2006), the expansion of sea ice may be associated with a cooling triggered by a negative solar irradiance anomaly,
which was amplified through a positive oceanic feedback mechanism. The cooling caused temporary relocation of deep-water
formation sites in the Nordic Seas, which was accompanied by a distinct reduction in AMOC strength (Hall et al., 2004). The
increase in sea-ice extent stratified the water column and hampered the deep-water formation, leading to additional cooling and
more sea ice (Renssen et al., 2006).

### 5.4 Variations of Atlantic and Polar Water entrainment in the EGC during the Holocene and their implications for the
general climatic trends in the Nordic Seas

Distinct variability in subsurface water mass properties that we see on the NE Greenland shelf through the Holocene point to broad-
scale changes in the proportion of RAW/AAW and PW in the EGC, and thus, to a strong covariance with the northward heat
transport in the Nordic Seas (e.g., Moros et al., 2012). The increased entrainment of warm Atlantic-sourced water into the EGC
that we document during the HTM (ca. 8.2-6.2 ka BP at our study site), coincides with increased advection of warm AW in the
West Spitsbergen Current, driven by increased wind force and/or by stronger thermohaline circulation (Sarthein et al., 2003;
Slubowska-Woldengen et al., 2007; Knudsen et al., 2011; Werner et al., 2013; Werner et al., 2016; Consolaro et al., 2018).

During the Mid Holocene (8.2-4.2 ka BP; Walker et al., 2018), the summer insolation declined at the northern latitudes due to the
changes in the Earth´s orbital parameter (Berger, 1978). From 7 ka BP, the summer sea surface temperatures of the Nordic Seas
started to decrease (Koç et al., 1993; Andersen et al., 2004). This cooled AW progressed northward carrying less and less heat to
the North Atlantic and West Spitsbergen currents (Sarthein et al., 2003; Slubowska-Woldengen et al., 2007; Risebrobakken et al.,
2011; Consolaro et al., 2018; Hald et al., 2007), also due to a weakened AMOC (e.g., Bauch et al., 2001; Hall et al., 2004; Hald et
al., 2007; Knudsen et al., 2011). We suggest that, in response to the reduced warm water transport through the RAC, the proportion
of cooled AAW and PW in the EGC increased, which ended the ameliorated conditions on the East Greenland shelf seen at our
study site.

The northward flow of AW with the North Atlantic and West Spitsbergen currents to the eastern Fram Strait arrived to a minimum
during the late Holocene, after 4.5 ka BP (Slubowska-Woldengen et al., 2007). Assuming that the fraction of AW entering the
Arctic Ocean and the RAC stayed constant, the RAW reaching the EGC would also have weakened. Consequently, the EGC
became stronger, carrying a thick layer of fresh PW southward, as also seen after ca. 4.2 ka BP (and even more so after 3.2 ka BP)
in DA17-NG-ST07-73G. A similar pronounced colder period after 4 ka BP is recognised in some (but not all) reconstructions north
and south of Iceland, in the Greenland Sea and to the west of Svalbard (Moossen et al., 2015; Orme et al., 2018; Telesinski et al.,
2014b; Werner et al., 2016; respectively); in particular at locations influenced by the EGC. During intervals of a negative NAO/AO
high pressure over Greenland and consequently stronger northerly winds over East Greenland can lead to strengthened outflow of
PW from the Arctic Ocean through the Fram Strait (e.g., Hurrell et al., 2003). This PW will be carried southward by the EGC (e.g.,
Ionita et al, 2016). We therefore associate the gradual cooling trend seen in our record as well as at other North Atlantic sites with
a continuous transition from a more prevailing positive NAO/AO phase towards a period of more negative NAO/AO(e.g., Andersen
et al., 2004; Orme et al., 2018).; The late-Holocene trend towards a stronger and fresher EGC is likely to have increasingly impacted

the Subpolar Gyre dynamics (e.g., Born and Stocker, 2014). Increased freshwater input to the Labrador Sea may have weakened the Subpolar Gyre circulation by preventing deep convection (Hillaire-Marcel et al., 2001), and consequently reduced the amount of water entrained into the North Atlantic Current (Moros et al., 2012). This, on the other hand, may have forced enhanced northward heat transport via the North Atlantic Current, and may have helped to restart (Thornalley et al., 2009) the previously weakened AMOC (e.g., Cronin et al., 2003; Oppo et al., 2003).

## 6 Conclusions

The presented multiproxy study, based on benthic foraminiferal assemblage-, geochemical- and sedimentological analyses of sediment core DA17-NG-ST07-73G allowed us to reconstruct changes in sea surface productivity, subsurface water temperatures, sea-ice conditions, Greenland Ice Sheet melting and thus, in the strength of the EGC over the last ca. 9.4 ka BP on the NE Greenland shelf.

-   Between 9.4 and 8.2 ka BP the water column was highly stratified, with cold, heavily sea ice-loaded surface waters and strong influx of warm waters in the subsurface. High surface water productivity suggests that the polar front was close to our location.
-   A short-lived peak in terrestrially-derived elements suggesting transport of sediments from land via meltwater or icebergs may be linked to the so-called 8.2 ka BP event.
-   The interval from 8.2 to 6.2 ka BP was characterised by the warmest bottom-water conditions of the Holocene on the NE Greenland shelf, with low surface water productivity and strong Atlantic Water (AW) influence from a persistent influx of RAW (from the RAC) to a generally weakened EGC.
-   After the HTM, the water column started to again resemble that of the period prior to 8.2 ka BP. The EGC became stronger, with sea ice-loaded surface waters and relatively warm Atlantic-sourced subsurface waters. The subsurface inflow of warm AW from the RAC decreased, but remained persistent.
-   After 4.2 ka BP increased Polar Water (PW) at the surface of the EGC and reduction of the RAW at subsurface levels led to freshening and reduced stratification of the water column and to a (near) perennial sea-ice cover.
-   The period from ca. 3.2 until 0.3 ka BP is characterised by cold and unstable conditions, with minimum surface water productivity, and possibly, sea-ice cover at our location. During this period, the EGC was likely strengthened, with a thick layer of cold and fresh PW on the surface and strong recirculated AAW inflow from the Arctic at subsurface levels.
-   The proportion of PW and AW in the EGC shows a strong covariance with the northward heat transport to the northern North Atlantic. Thus, the cooling trend that characterises the Holocene after the HTM led to strengthened EGC and increased PW transport, probably due to transition to a more negative NAO/AO scenario, causing more PW to exit the Arctic Ocean via the Fram Strait. This stronger and fresher EGC in the Late Holocene likely impacted the Subpolar Gyre circulation and helped strengthen the AMOC.

## Appendices

**Table A1: Comparison of radiocarbon dates calibrated with the Marine20 (Heaton et al., 2020) and the Marine13 (Reimer et al., 2013) calibration curve in core DA17-NG-ST07-73G. For the dates calibrated with the Marine20 we used a local marine reservoir age correction (ΔR) of 0 ± 50 years; for the dates calibrated with the Marine13 we used a ΔR of 150 ± 50 years, as this is the approximate offset between the two curves (Heaton et al., 2020).**

| Lab. ID | Depth (cm) | Radiocarbon age (yr BP) | Error (yr) | Median cal. age (yr BP) Marine20, ΔR=0±50 yr | Median cal. age (yr BP) Marine13, ΔR=150±50 yr |
|---|---|---|---|---|---|
| ETH-95387 | 70.5 | 2475 | 60 | 1952 | 1946 |
| ETH-95388 | 110.5 | 3275 | 70 | 2935 | 2915 |
| ETH-110893 | 226 | 5645 | 30 | 5835 | 5872 |
| ETH-95389 | 250.5 | 6015 | 70 | 6241 | 6284 |
| ETH-95390 | 300.5 | 7595 | 70 | 7870 | 7902 |
| ETH-95391 | 345.5 | 9015 | 70 | 9536 | 9528 |

**Table A2: Foraminiferal taxa identified in this study (core depth 345-0 cm), with their original reference. Benthic calcareous, agglutinated and planktic species are separated and listed alphabetically.**

**Benthic agglutinated**

*Adercotryma glomerata* (Brady, 1878)

*Ammodiscus* sp.

*Ammoglobogerina globigeriniformis* (Parker and Jones, 1865)

*Cibrostomoides kosterensis* (Höglund, 1947)

*Deuterammina grahami* Brönnimann and Whittaker, 1988

*Deuterammina montagui* Brönnimann and Whittaker, 1988

*Hormosinella* sp.

*Portatrochammina bipolaris* Brönnimann and Whittaker, 1980

*Recurvoides turbinatus* (Brady, 1881)

*Reophax fusiformis* (Williamson, 1858)

*Reophax guttifer/rostrata* (Brady, 1881)/Höglund, 1947

*Reophax* sp.

*Reophax subfusiformis* Earland, 1933

*Rhabdammina* sp.

*Saccammina difflugiformis* (Brady, 1879 )

*Saccammina sphaerica* Brady, 1871

*Saccorhiza ramosa* (Brady, 1879)

*Spiroplectammina biformis* (Parker and Jones, 1865)

*Textularia* cf. *earlandi* Parker, 1952

*Textularia torquata* Parker, 1952

*Texularia kattagatensis* Höglund, 1948

**Benthic calcareous**

*Islandiella helanae* Feyling-Hanssen and Buzas, 1976

*Ammonia* sp.

*Astrononion gallowayi* Loeblich and Tappan, 1953

*Bolivina* aff. *albatrossi* Cushman, 1922

*Buccella frigida* (Cushman, 1922)

*Buliminella elegantissima* (d´Orbigny, 1839)

*Cassidulina neoteretis* Seidenkrantz, 1995

*Cassidulina reniforme* Nørvang, 1945

*Ceratobulimina arctica* Green, 1959

*Cibicides lobatulus* (Walker and Jacob, 1798)

*Dentalina pauperata* d´Orbigny, 1846

*Elphidium albiumbilicatum* (Weiss, 1964)

*Elphidium clavatum* Cushman, 1930

*Elphidium frigidum* Cushman, 1933

*Elphidium hallandense* Brotzen, 1943

*Eoponidella pulchella* (Parker, 1952)

*Epistominella arctica* Green, 1959

*Epistominella vitrea* Parker, 1953

*Fisurina* sp.

*Florius* sp.

*Glabratella arctica* Scott and Vilks, 1991

*Globulina oculus* Jennings, Seidenkrantz and Knudsen, 2020

*Guttulina glacialis* (Cushman and Ozawa, 1930)

*Haynesina nivea* (Lafrenz, 1963)

*Islandiella norcrossi* (Cushman, 1933)

*Lagena* sp./*Procerolagena*

*Melonis barleeanum* (Williamson, 1858)

*Miliolinella subrotunda* (Montague, 1803)

*Nonionella iridea* Heron-Allen and Earland, 1932

*Nonionella labradorica* (Dawson, 1860)

*Polymorphina* sp.

*Pullenia bulloides* (d´Orbigny, 1846)

*Pyrgo williamsoni* (Silvestri, 1923)

*Quinqueloculina seminulum* (Linnaeus, 1758)

*Quniqueloculina stalkeri* Loeblich and Tappan, 1953

*Robertina arctica* d´Orbigny, 1846

*Sagrina* sp.

*Stainforthia concava* (Höglund, 1947)

*Stainforthia feylingi* Knudsen and Seidenkrantz, 1994

*Stainforthia fusiformis* (Williamson, 1858)

*Stetsonia horvathi* Green, 1959

*Trifarina fluens* (Todd in Cushman and McCulloch, 1948)

*Triloculina tricarinata* d´Orbigny, 1846

*Triloculina trihedra* Loebich and Tappan, 1953

**Planktic**

*Neogloboquadrina incompta* (Cifelli, 1961)

*Neogloboquadrina pachyderma* (Ehrenberg, 1861)

*Turborotalita quinqueloba* (Natland, 1938)

**Table A3: List of benthic foraminiferal key species used for palaeoenvironmental reconstruction.**

| Species | Environmental preferences | References |
|---|---|---|
| Calcareous | | |
| *Cassidulina neoteretis* | warm, saline Atlantic Water | Mackensen and Hald, 1988; Seidenkrantz, 1995; Rytter et al., 2002; Jennings et al., 2004; Jennigns et al., 2011, Perner et al., 2015; Cage et al., 2021 |
| *Pullenia bulloides* | warm, saline Atlantic Water | Rytter et al., 2002; Jennings et al., 2011 |

| | | |
|---|---|---|
| *Cassidulina reniforme* | chilled Atlantic Water, stable conditions | Hald and Korsun, 1997; Slubowska et al., 2005; Slubowska-Woldengen et al., 2007; Perner et al., 2011; Consolaro et al., 2018 |
| *Islandiella norcrossi* | chilled Atlantic Water | Slubowska-Woldengen et al., 2007; Perner et al., 2011; Perner et al., 2015; Cage et al., 2021 |
| *Melonis barleeanum* | productivity, cool Atlantic Water | Perner et al., 2015 |
| *Elphidium clavatum* | tolerates unstable conditions, in glaciomarine environments | Slubowska-Woldengen et al, 2007; Jennings et al., 2011; Perner et al., 2011 |
| *Stetsonia horvathi* | cold, Arctic deep water, extensive sea ice | Green, 1960; Lagoe, 1979; Wollenburg and Mackensen, 1988; Jennings et al., 2020 |
| *Epistominella arctica* | cold, Arctic deep water, productivity, sea ice (edge) | Green, 1960; Lagoe, 1979; Wollenburg and Mackensen, 1988; Jennings et al., 2020 |
| *Stainforthia feylingi* | tolerates unstable conditions, high productivity, often in sea-ice edge regions | Knudsen and Seidenkrantz, 1994 |
| Agglutinated | | |
| *Adercotryma glomerata* | Atlantic sourced waters | Hald and Korsun, 1997, Lloyd 2006; Perner et al., 2012; Wangner et al., 2018 |
| *Textularia earlandi* | cold, low-salinity Arctic waters | Jennings and Helgadottir, 1994; Korsun and Hald, 2000; Wangner et al., 2018 |
| *Textularia torquata* | cold, low-salinity Arctic waters | Perner et al., 2012; Perner et al., 2015; Wangner et al., 2018 |
| *Spiroplectammina biformis* | cold, low-salinity Arctic waters | Jennings and Helgadottir, 1994; Korsun and Hald, 2000; Perner et al., 2012; Wangner et al., 2018 |
| *Saccammina difflugiformis* | Polar Waters on the EG shelf | Jennings and Helgadottir, 1994 |

**Table A4: List of abbreviations used in the manuscript.**

| | |
|---|---|
| NE Greenland | North East Greenland |
| EGC | East Greenland Current |
| RAC | Return Atlantic Current |
| AMOC | Atlantic Meridional Overturning Circulation |
| NAO | North Atlantic Oscillation |
| AO | Arctic Oscillation |
| PW | Polar Water |
| RAW | Return Atlantic Water |
| AAW | Arctic Atlantic Water |
| AW | Atlantic Water |
| HTM | Holocene Thermal Maximum |

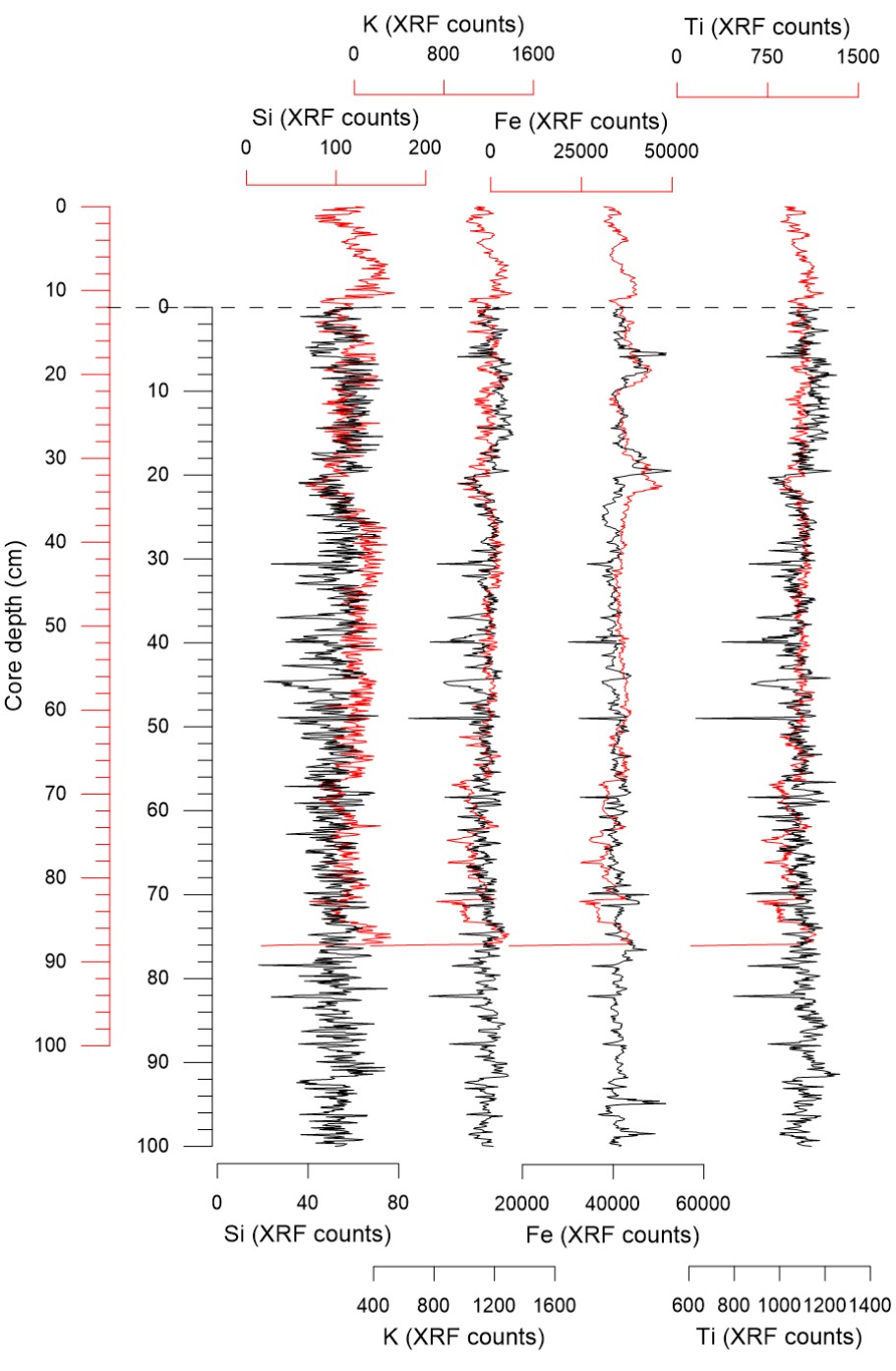

Figure A1: Comparison of XRF data from gravity core DA17-NG-ST07-73G (black) and Rumohr core DA17-NG-ST07-72R (red).

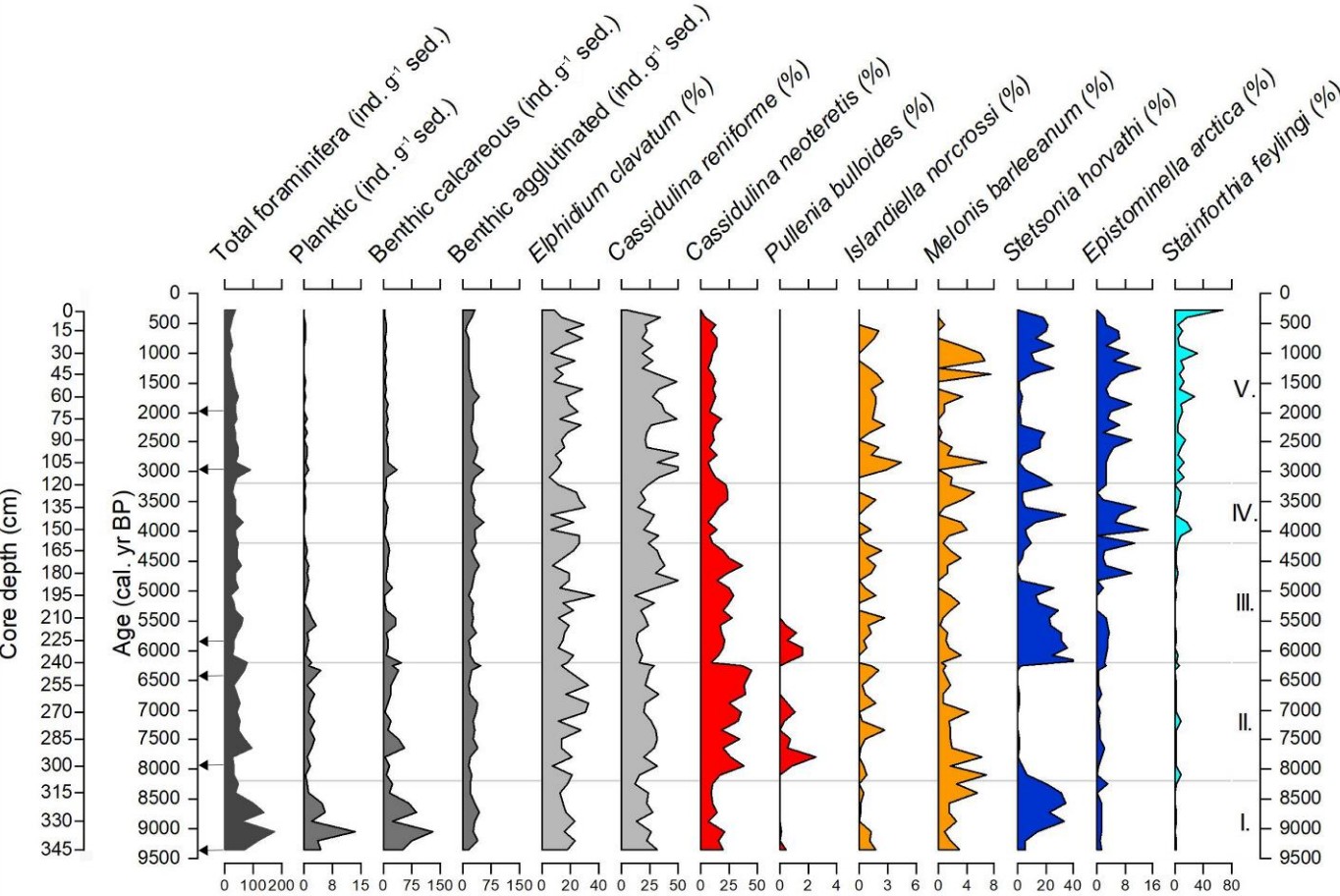

**Figure A2: Foraminiferal concentrations and relative abundances of nine selected calcareous benthic species (expressed as a percentage of total benthic calcareous foraminiferal content) versus calibrated age along sediment core DA17-NG-ST07-73G. The depicted species were chosen in order to show changes in the environment. Red colour represents species that indicate warm Atlantic Water inflow; orange represents species that indicate in the Arctic recirculated, chilled Atlantic Water influence; dark blue represents Artic Water species; light blue represents a species that indicates sea ice. Ecozones (I-V) are shown on the right side of the figure. Black arrows next to the left primary y-axis (Age) mark radiocarbon dates. Note that the x-axes have different scaling.**

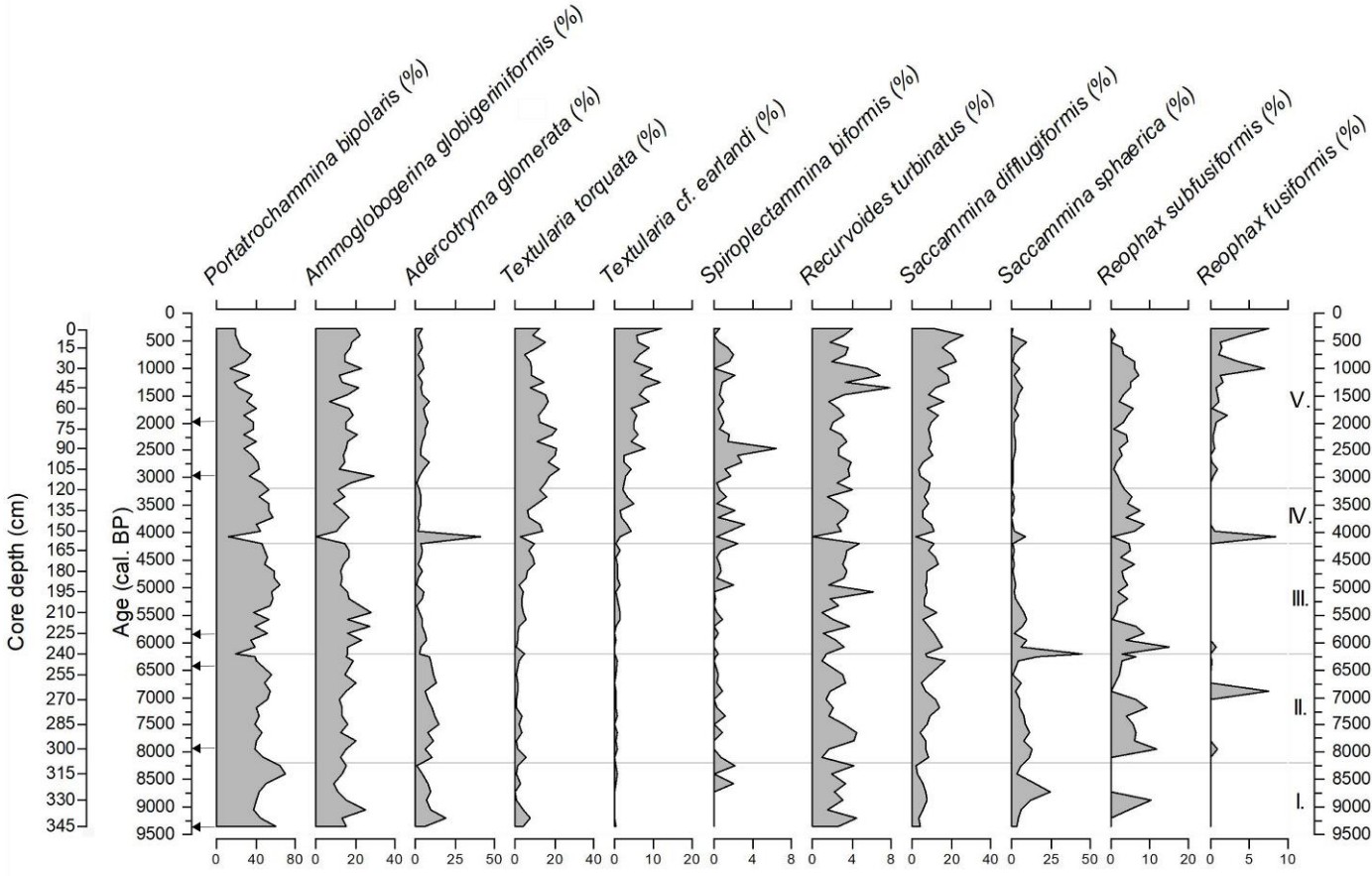

**Figure A3: Relative abundances of the most abundant (>5% in at least one sample) agglutinated benthic foraminifera species (expressed as a percentage of total benthic agglutinated foraminiferal content) versus calibrated age along sediment core DA17-NG-ST07-73G. Ecozones (I-V) are shown on the right side of the figure. Black arrows next to the left primary y-axis (Age) mark radiocarbon dates. Note that the x-axes have different scaling.**

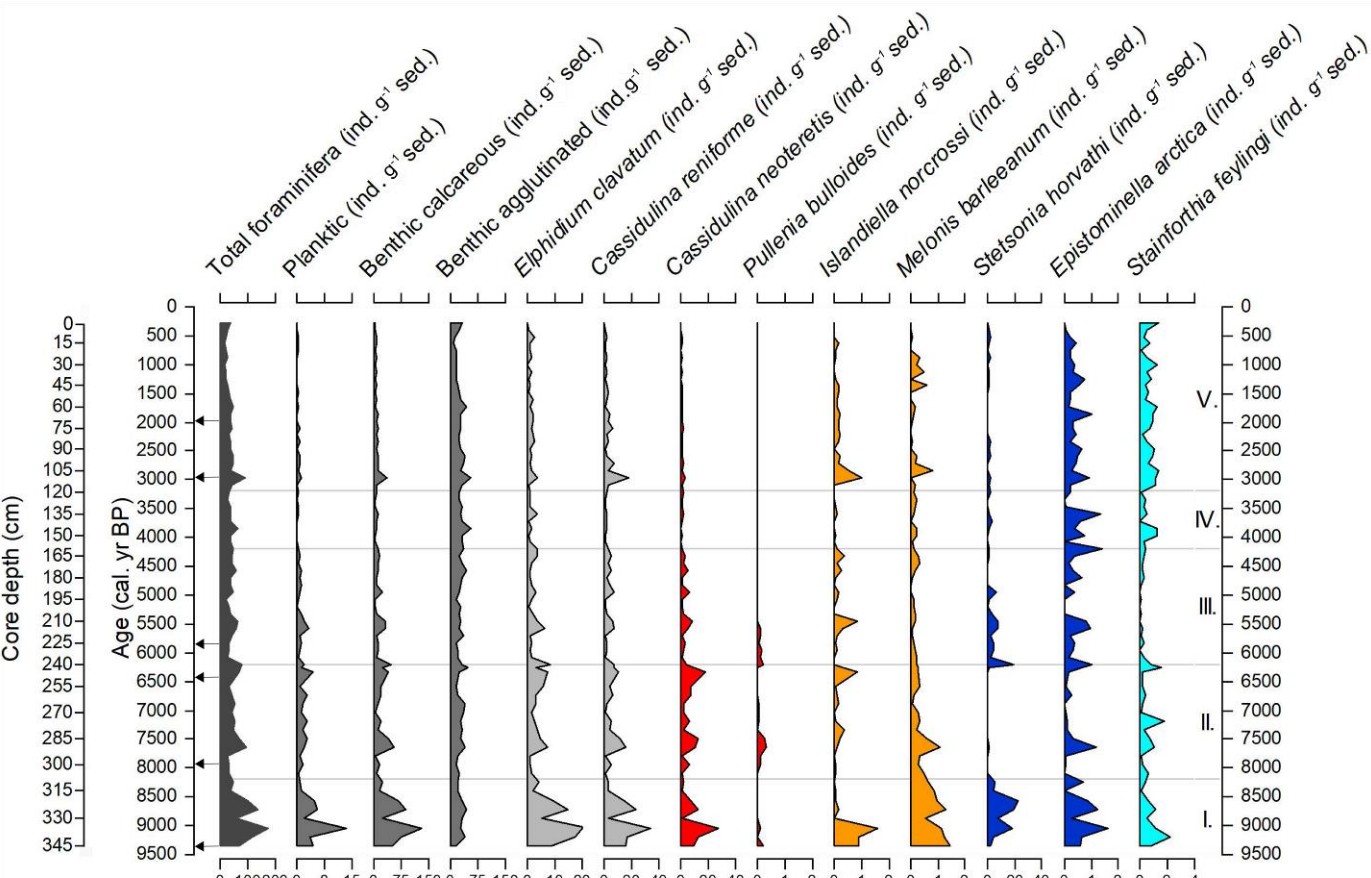

**Figure A4:** Foraminiferal concentrations showing the concentrations of nine selected calcareous benthic species versus calibrated age along sediment core DA17-NG-ST07-73G. The depicted species were chosen in order to show changes in the environment. Red colour represents species that indicate warm Atlantic Water inflow; orange represents species that indicate in the Arctic recirculated, chilled Atlantic Water influence; dark blue represents Artic Water species; light blue represents a species that indicates sea ice. Ecozones (I-V) are shown on the right side of the figure. Black arrows next to the left primary y-axis (Age) mark radiocarbon dates. Note that the x-axes have different scaling.

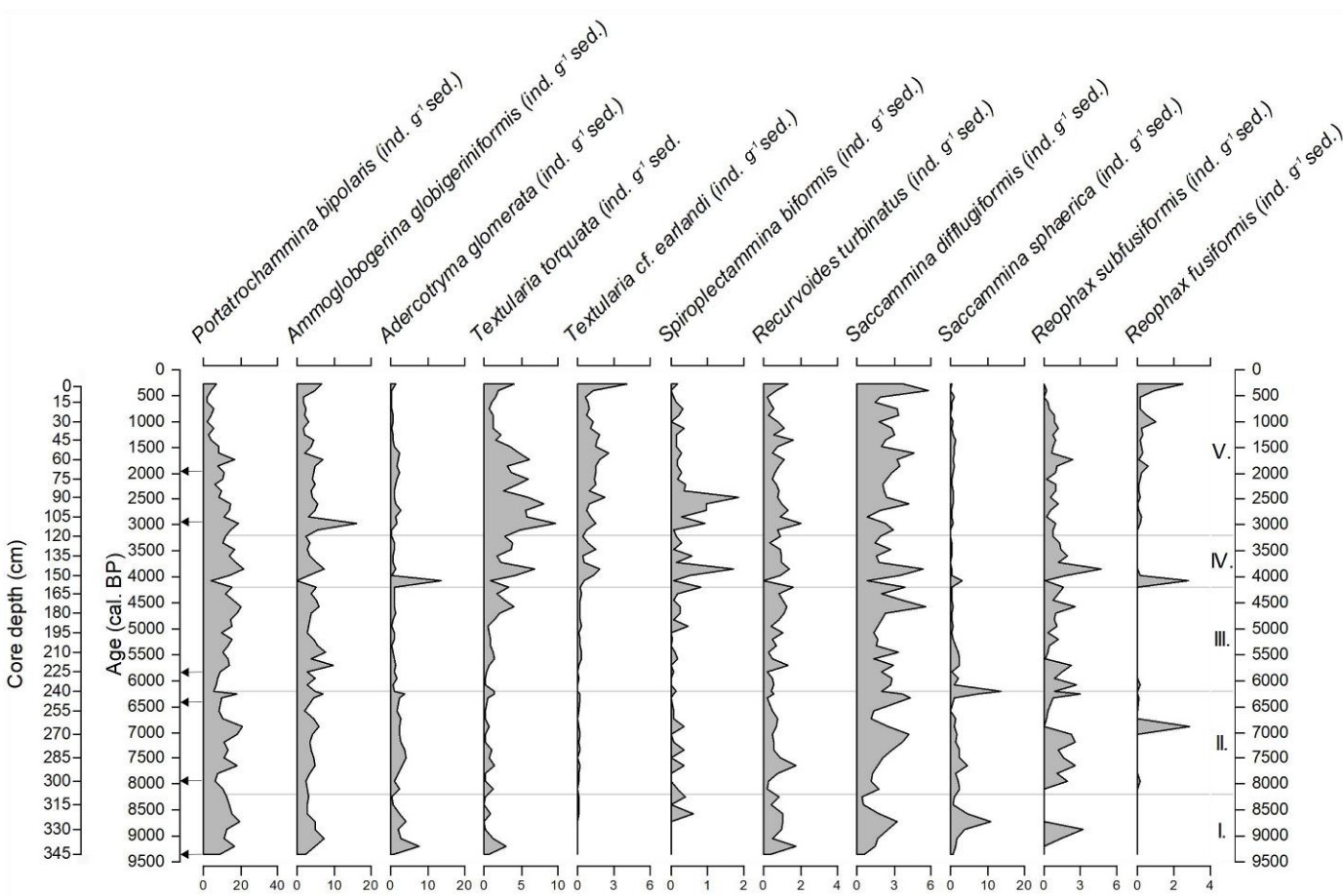

**Figure A5: Foraminiferal concentrations of the most abundant (>5% in at least one sample) agglutinated benthic foraminifera species versus calibrated age along sediment core DA17-NG-ST07-73G. Ecozones (I-V) are shown on the right side of the figure. Black arrows next to the left primary y-axis (Age) mark radiocarbon dates. Note that the x-axes have different scaling.**

### Data availability

Data presented in this manuscript have been uploaded to PANGEA (https://doi.pangaea.de/10.1594/PANGAEA.934100) and will be publicly accessible right after the publication of the manuscript.

### Author contribution

Teodora Pados-Dibattista and Marit-Solveig Seidenkrantz developed the research idea. Teodora Pados-Dibattista carried out the sampling, data collection and the data analysis, with the help of Christof Pearce, Henrieka Detlef, Jørgen Brendtsen and Marit-Solveig Seidenkrantz. Christof Pearce performed the age modelling of the core. Teodora Pados-Dibattista prepared the manuscript with contributions from all co-authors.

### Competing interests

Author Marit-Solveig Seidenkrantz is co-editor-in-chief of the journal.

**Acknowledgements**

We would like to thank the captain and crew as well as the shipboard scientific party onboard RV *Dana*. We also wish to thank Marianne Lyngholm Nielsen for carrying out the X-Ray Fluorescence core scanning and the magnetic susceptibility measurements and Nils Andersen, Leibniz Laboratory for Radiometric Dating and Stable Isotope Research at the Christian-Albrechts-University of Kiel, for the stable isotope measurements.

**Financial support**

The NorthGreen17 expedition was funded by the Danish Centre for Marine Research and the Natural Science and Engineering Research Council of Canada. The research was funded by the European Union´s Horizon 2020 research and innovation programme under grant agreement No. 792639, with further support from the Danish Council for Independent Research (grants no. 7014- 00113B (G-Ice) and 0135-00165B (GreenShelf) to MSS)) and the European Union's Horizon 2020 research and innovation program under grant agreement No. 869383 (ECOTIP).

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
