# Peer review of "Holocene palaeoceanography of the Northeast Greenland shelf"

_Climate of the Past, 2021_

## Author Comment (AC1)

Referee 1

Review of "Holocene paleoceanography of the Northeast Greenland shelf" by Teodora Pados-Dibattista, Christof Pearce, Henrieka Detlef1, Jørgen Brendtsen, Marit-Solveig Seidenkrantz.

This manuscript is an interesting contribution to a number of ongoing international efforts to investigate the region of and off NE Greenland. This region is of particular importance for the Greenland Ice Sheet because it holds the NE Greenland Ice Stream, which accounts for a large part of the ice export from the ice sheet. The manuscript builds on established foraminiferal, isotopic and sediment-chemical data sets from a long sediment core obtained on the NE Greenland shelf which allow to reconstruct environmental change in this area at high temporal resolution and in great detail. The results are novel, the data interpretation is well-founded and in general I am in favor of a publication of this data set in Climate of the Past.

We are grateful to the reviewer for their valuable comments, which have been important for improving the manuscript. We have done our best to make corrections to the manuscript accordingly. In a few instances, we have chosen a different solution to an issue pointed out by the reviewer; in those cases, we provide explanations to this under each comment below.

However, I have two major concerns which should be addressed by the authors before publication is possible:

(1) The age model is based on a calibration which uses the Marine20 data set of Heaton et al. (2020, Radiocarbon). In this paper, Heaton et al. explicitly state already in the abstract (and in detail in the text) that the Marine20 data set "is not suitable for calibration in polar regions". Accordingly, Pados-Dibattista et al. need to find alternative ways of calibrating their radiocarbon data. They may think of using the IntCal20 data set and a suitable local reservoir correction. Proposed corrections have been published by e.g.,

Tauber, H., Funder, S., 1975. 14C content of recent molluscs from Scoresby Sund, central East Greenland. Grønlands Geol. Unders. Rapp. 75, 95–99.

Mangerud, J., Bondevik, S., Gulliksen, S., Hufthammer, A.K., Høisæter, T., 2006. Marine 14C reservoir ages for 19th century whales and molluscs from the North Atlantic. Quat. Sci. Rev. 25, 3228–3245.

Coulthard, R.D., Furze, M.F.A., Pienkowski, A.J., Nixon, F.C., England, J.E., 2010. New marine ΔR values for Arctic Canada. Quat. Geochronol. 5 (4), 419–434).

Thank you for your comment. The reviewer is correct that Heaton et al. (2020) state that the Marine20 is not suitable for polar regions. The same was true, however, for Marine13; it was just not explicitly stated as for the latter calibration curve. The presence of sea ice in polar regions impacts the local reservoir age, and therefore there is added uncertainty; this issue is not resolved by using an older calibration dataset. It is also not resolved by using the terrestrial IntCal20 as this would lead to further uncertainties. Without the presence of alternative dating methods (e.g. tephrochronology, paleomagnetism), all we can do is acknowledge this added uncertainty in the chronology of these Arctic marine sediment archives. Moreover, since its publication, the Marine20 has been widely used in the Arctic

realm, e.g., Farmer et al., 2021 (Nat. Geoscience https://doi.org/10.1038/s41561-021-00789-y), Altuna et al., 2021 (Commun. Earth Environment https://doi.org/10.1038/s43247-021-00264-x) to name just a few. Finally, the differences for this specific Holocene reconstruction between using Marine13 and Marine20 are much smaller than the associated uncertainties. When evaluating the reservoir age we have taken all existing literature into account.

(2) The Discussion chapter needs to be reorganized. Currently, in its first part it consists of several subchapters (6.1-6.2) discussing Holocene environmental change on the NE Greenland shelf as derived from own data. This text is mostly fine, but more comparisons should be made with the paper of Zehnich et al. (2020) which contains (among other data) benthic isotopic data sets of higher temporal resolution than the ones of Pados-Dibattista et al. In the present manuscript the second part of the Discussion chapter holds two subchapters (6.3 and 6.4) which present a review of published knowledge concerning larger-scale Holocene climatic and environmental connections in and around the research area. What is missing is the combination of both parts. The authors need to show how their own results relate to larger scale developments and how they may improve our understanding of these developments.

Each paragraph in the discussion chapter 5.3 (Paleoenvironmental interpretation) starts with a short environmental interpretation of the data from our own core. It is necessary first to provide an environmental interpretation before discussing its significance. We chose to combine this environmental interpretation with the broader discussion in order to both avoid repetitions and make the links clearer. However, we keep the short interpretation in a separate subsection, in order to clearly separate, which part of the discussion is based on our new study, and which is based on comparison to previous studies.

The last two paragraphs of the discussion (5.4 and 5.5) place our results in an even broader context, but here we have now added more references to our own data. Moreover, we have added more comparisons to Zehnich et al. (2020).

Minor and more specific comments and proposed corrections (general and by line numbers):

Check the entire manuscript for consistency:

- sea ice vs. sea-ice
Corrected.

- West Spitzbergen Current vs. West Spitsbergen Current
Corrected.

Be consistent with using either British or American spelling (grey/gray, colour/color, -ise/-ize)
Done.

9: stable isotope and
Corrected.

14: iceberg
Corrected.

26-27: The reader might want to know why the "societal and environmental relevance of this sea-ice reduction" is particular important for Greenland...
Sentence added.

34: meltwater
Corrected.

35: budget and stratification, and it influences
Corrected.

36: Indicate Northeast Greenland ice stream on the map! Later you use "Northeast Greenland Ice Stream". Be consistent with capital letters!
Corrected and added text with the glacier outlets. We indicated the Northeast Greenland Ice Stream on the map (Fig. 1).

39ff: Is it necessary to mention all the site numbers and citations in the figure caption if they reappear in the Discussion anyway? I guess something like "Locations of cores discussed in the text are indicated" would be enough...
Thank you for your comment, however, we think that it is quite important for the easy overview to indicate the site numbers and citations in the figure capture. In this way, the reader doesn´t have to spend a lot of time looking for this information in the text and can compare easily our results to other relevant papers.

51: Moossen
Corrected.

55/56: "the returning branch of the West Spitzbergen Current" - if you mean the RAC, then call it RAC!
Corrected.

57: freshwater (check also in the entire manuscript!)
Corrected.

58: affects
Corrected.

63: Better: demanding an improved...
Corrected.

65-69: You should either be more specific in explaining the features connected with the NAO or delete this paragraph and introduce NAO later. As it reads now, it is very general and details (e.g., NAO+ and NAO-) need to be introduced later, anyway (e.g., "a redistribution of atmospheric mass" - what kind?; shifts from one phase to another" - what kind of "phases"?)
We agree that the description was rather generic and we have thus moved the detailed description of NAO (which describes the impact on Atlantic Water inflow in the Fram Strait and sea-ice formation) from the discussion to the introduction.

76: most of the Holocene
Corrected.

82: of the NE
Corrected.

83: better "neighbored banks"? The banks are not really surrounding the troughs!
Thank you for your comment. We do see your point, but we took the term from Arndt et al., 2015, which is describing the detailed bathymetry of the Northeast Greenland continental shelf. As this paper has become a standard background paper for the region, we prefer to keep this terminology.

87: Johannessen (check also in ref list!)
Corrected.

88-89: In your Fig. 2a, waters with S<32 only reach down to 150 m!
In figure 2a we show CTD data, which is reflecting the water column at the moment of sampling. On 2b we show an annual temperature average. We added this information to the figure capture to make the difference more understandable.

90: Atlantic sources
Corrected.

93-94: From Fig. 1 I cannot see that the RAC runs along the Greenland coast.
We changed the text in order to make it more understandable that the RAC is joining the EGC.

99: modulates the glaciers' basal
Corrected.

101-102: Make two sentences!
Corrected.

102: Start sentence with "Today, ..."
Corrected.

103: Polar Front lie east...   Start new sentence with "However, ..."
Corrected.

105-107: Make two sentences!
Corrected.

108: increases in size
Corrected.

108-110: Make two sentences!
Corrected.

113 vs. 120: Why are coordinates of the core site differing in detail?
We added "coring station" in the figure capture of figure 2, in order to indicate that the CTD sampler and the gravity corer was deployed at slightly different positions but at the same station.

117: Temperatures (WOA) from which season?
Added "Annual average".

117: this transect
Corrected.

140: top sediment loss?
Corrected.

155: are shown
Corrected.

159: the 100-1000 µm fraction
Corrected.

167: bulk sample
Corrected.

169: intervals
Corrected.

170: The official silt size is 2-63 microns. Are you sure that you used a 60 micron mesh?
The particle sizes measured are dependent on the available instrument. We did not use sieves
(i.e. thus not a 63 µm mesh) to identify the particle sizes but a laser diffractometer. The laser
particle sizer (Sympatec Helos) at the Department of Geoscience, AU has settings for these
three groups: sand (>60 µm), silt (2-60 µm) and clay (<2 µm), and the 60 µm is so close to
the 63 µm that this does not have any significant impact on the fractions.

176: Clearly describe what is shown in this figure, from left to right!
Corrected.

185: Better: below the lowest radiocarbon-dated sample
Corrected.

186-187: Give a cross-reference to chapter 4.4 for the reworked species.
Corrected.

191: ... and focus on the last 9.4 ka.
Corrected.

200: lowest radiocarbon-dated level
As we modified slightly the figure (instead of dashed line a hatched box under the lowest
radiocarbon date), we exchanged this sentence to: "The hatched box in the bottom of the core
indicates an interval containing reworked microfossils and is therefore of uncertain age".

203: insert reference to Fig. 3
Inserted.

203-204: I agree that there is mostly a good visual correlation (maybe you should calculate correlation coefficients?), but I do not see a "trend" to either higher or lower values. If it is there, it is weakly developed.

We have changed the text to "relatively constant values".

206: For me this "steady increase" is hard to see.

The reviewer is right, we have reformulated the sentence to: "followed by a steady increase until 125 cm (ca. 3.2 ka BP) and rather constant values until the top of the core". Moreover, we added a mean-line to all four XRF curves in figure 3, in order to make it easier for the reader to see the described changes.

207: I think the variability in Ca/Fe between the core base and c. 340 cm is more than just a "slightly stronger fluctuation". The amplitude is orders of magnitude higher than in the rest of the core!

The reviewer is right about that this sentence was not well formulated. We added a sentence at the beginning of this section to point out that later we are only describing the last 345 cm of the core. Moreover, we added "from 345 cm until ~270 cm" to the mentioned sentence, in order to emphasize that we do not describe the very end of the core here.

212-218: Be more specific and clearly distinguish between modern and extinct species!

Done.

229 and 236: I do not think that it makes sense to give an average percentage in the entire core for the total agglutinated or total calcareous species, especially if (as you write) the relative proportion is changing from the core base to the top. I ask you to calculate % agglutinated of all benthics for each sample and add this record to Fig. 5, also because you refer several times to the aggl/calc ratio later in the text.

We agree that when evaluating percentage data, it needs to be taken into account that the number of agglutinate specimens decrease downwards. This is also, why we have shown the relative frequencies vs. only agglutinated specimens and calcareous only vs. calcareous in figures in the supplementary. However, we now also show the % agglutinated of the total benthic foraminiferal assemblage in Fig. 5. Moreover, in response to a comment by reviewer 2, we have added two figures showing the concentrations (individuals of species per gram sediment) to the supplementary (Fig. A4 and A5).

230: Start new sentence with "However, ..."

Corrected.

238: Again, I cannot see a "steadily decreasing trend towards the top of the core". Values are relatively high near the core base, around 6.5 ka, and near the top. In between they are lower. No trend is visible...

Corrected.

244: Arctic

Corrected.

245: Why is there a period (.) behind the Roman letters for the ecozones? Looks strange...

Corrected.

245-246: It is more common to say "horizonal" and "vertical" axes.

Thank you for your comment. However, after consulting with two English native speakers, we believe that x and y axes are the right terms. We would like to ask the Editor to advise us, whether the journal has some preferences on this topic.

257: Once "on average" is enough…
We agree that the repetition of "on average" seems somewhat obsolete and irritating. However, as we worry that the information could otherwise be misleading or raise questions, we have added the following sentence prior to the foraminiferal zone descriptions: "Unless otherwise specified all relative frequencies are provided as average values for the interval."

294: A trend means that values are changing in one direction, i.e., they become higher or lower. If values are mostly the same, then there is no trend. One would rather say that values remain constant (within a certain range).
Corrected.

298: Wouldn't "cluster" be the proper term?
Thank you for your comment, however, we worry that the term cluster could be misunderstood as derived from cluster analyses. The term "groups" or "groupings" is in fact very commonly used in this context (e.g., Rasmussen and Thomsen, 2004; Perner et al., 2012; Seidenkrantz et al., 2021). Thus, we believe that group is the term that is more fitting to our description.

308: in our interpretation
Corrected.

313: and in fjords
Corrected.

315: I suggest to label the panels/records (a) to (i) and give these labels in the figure caption together with the description of the individual panels/records. This will make it easier for the reader to identify certain records.
Good point, we have added labels.

325-326: How can the bottom waters in these troughs be distinguished?
According to e.g., Budéus et al. (1997) and Schaffer et al. (2017) Atlantic water masses on the bottom of Norske and Westwind Trough differ in their temperature and salinity. We added a sentence in order to explain this topic better.

327: but may also be present due to...
Corrected.

328: shelf break
Corrected.

334-337: Kapp København is a location, not a deposit. Where are such places with Pliocene/early Pleistocene sediments? I cannot find anything on the map (Fig. 1). Is it likely that sediments were transported to your site, and how? Writing that "the breakup and significant retreat of a nearby glacier caused reworking of older sediments" is too general and the example (Seidenkrantz et al. 2019) is from far in the south...

It is actually the Kap København Formation in Peary Land; we by mistake used an abbreviation. The paragraph has now been rewritten to better explain our suggestion, and the location is marked on Fig. 1b.

340: What is the evidence that this was a "cold interval"? You should avoid introducing such a-priori statements before you discuss your own (and published) paleoclimatic evidence. Corrected. The word "cold" was deleted.

344: "would have characterised" ... if...? Corrected through rephrasing.

340-348: Stratification and water masses on the NE Greenland shelf are also discussed by Zehnich et al. (2020). You should compare the results - here and in the other subchapters of the discussion. We have now included more comparisons with Zehnich et al. (2020) in the text.

343: Here you say that the area was heavily sea ice covered. Later (l.349 and in the discussion of the Syring et al. 2020 results) you state that planktic foraminifers were abundant and productivity was high. At first sight this sounds contradictory and needs a proper discussion. I am aware that this discussion comes when the relation to the ice margin is discussed, but you may from the beginning say that the results are only apparently contradictory. Thank you for your comment, we have reformulated this part and added a sentence about the apparent contradiction of the results.

348: A reference is needed when certain elements are ascribed to sources. Added.

348: I cannot see that the d18O values are particularly low in this section. A potential influence of temperature changes on d18O should be discussed. There is evidence for an enhanced advection of Atlantic Water to the NE Greenland margin (Bauch et al., 2001). As shown in several papers on the W Svalbard margin, this advected AW was relatively warm, even when compared to today, and likely it was still relatively warm when it reached the NE Greenland margin as the RAC. Accordingly, there may be a temperature influence on the isotopic signal. Moreover, how would meltwater (near the surface) influence the d18O of benthic organisms? We agree that there is no major difference between the d18O in this interval and the following intervals, but still, on average values are a bit lower. We have now added a short discussion on the potential causes, i.e. not just meltwater but also the impact of warmer bottom waters.

In any case, you should compare your isotope results (both d18O and d13C) to the isotope data sets of Zehnich et al. (2020). We have added more comparisons with Zehnich et al. (2020).

356ff: Try to find better arguments for a linkage of your event with the 8.2 ka event. Can you derive information on the nature of these sediments from the X-ray photos? What about grain sizes? Wouldn't more icebergs leave traces by IRD-rich sediments? Unfortunately, as mentioned in the text, we don´t have in the presented results other clues that would point to the 8.2 ka event. Our grain size analysis has a quite low resolution and it

doesn´t show any significant changes throughout the top 345 cm of the core. As mentioned in the text, the XRF peak could be only potentially linked to the 8.2 ka cold event. Further analyses are needed to be able to confirm this theory.

357ff: Long sentence. Split into two!
Corrected.

374: appeared around 8 ka BP in the record, after a long absence
Corrected.

378: This refers to the previous sentence and should not start a new paragraph.
Corrected.

384ff: Core numbers are not necessary here and in many other places when references are given.
Thank you for your comment, however, we believe that indicating the core numbers in the text makes it much easier for the reader to find the relevant core positions on Fig. 1.
Müller et al. discuss sea ice coverage and bioproduction, but temperatures only in a semiquantitative way. Werner et al. (2013, 2016) are more appropriate references for near-surface temperatures off W Svalbard and should be used here and in other subchapters of the discussion.
We exchanged Müller et al. 2012 to Werner et al. 2013 and 2016.
They show that strongest AW advection started c. 10.8 ka. On the other hand, Risebrobakken et al. (2011) showed that highest SSTs came only c. 9 ka. Since there is a strong influence of AW at the NE Greenland shelf seea floor, the timing of AW and temperature maxima should be discussed with reference to results from the E Fram Strait. Is there a discrepancy? Can you speculate why?
Thank you for your comment; however, we do discuss this topic later, in the discussion chapter 5.4, where we place our results into a broader context: "The drastic ice recession of the early Holocene produced an extended meltwater surface layer in the Greenland region prior to 8.6 ka BP (Seidenkrantz et al., 2013). The extensive melting of the Greenland Ice Sheet was strong enough to act as negative feedback to the early Holocene warming, and delayed the HTM with 2 kyr at our location compared to the eastern parts of the Nordic Seas (Blaschek and Renssen, 2013)."
One may also ask whether results from SE Greenland are suitable for comparisons. The Nordic Seas are much wider than the Fram Strait and the heat distribution by AW works in a different way there.
We believe that South East Greenland is an important location to mention and to compare to, as we are attempting to reconstruct the strength and the composition of the East Greenland Current, which flows south along the whole East Greenland coast.

403: Foster Bugt is not on the map in Fig. 1.
We changed Foster Bugt to Middle East Greenland shelf in the text.

402-406: Okay, this is interesting information. What kind of conclusion can you draw?
The conclusions of these changes recorded between 6.2 and 4.2 ka BP is mentioned in chapter 5.4, where we place our results in a broader context: "After the Thermal Maximum… The EGC became stronger, with sea-ice loaded surface waters and relatively warm Atlantic-sourced subsurface waters. Coincident with the expansion of the EGC, several studies from the Nordic Seas (e.g., Bauch et al., 2001; Hall et al., 2004; Hald et al., 2007) infer a

weakening of the AMOC, increased water column stratification and less ventilated subsurface during this period. In line with a decreased flux of recirculating AW onto the NE Greenland shelf, the Northeast Greenland Ice Margin started to advance from its mid Holocene minimum around 6 ka BP (Larsen et al., 2018)".

412: <1
Corrected.

414-415: Theoretically, high d18O could also result from stronger AW influence. You should build your arguments on a combination of proxy interpretations. Example: Since the forams point at low bottom water temperatures, we interpret the high d18O valus as indicative of...
Corrected.

420: 4.2 ka
Corrected.

427: What can you conclude from the literature information?
The conclusions of these changes recorded around 3.2 ka BP is mentioned in chapter 5.4, when we place our results in a broader context: "The Neoglacial cold interval started on the East Greenland shelf approx. 3.5-3.2 ka BP, with increased freshwater forcing from the Arctic Ocean and advance of the Greenland Ice Sheet (this study; Andersen et al., 2004; Jennings et al., 2011). According to model simulations of Renssen et al. (2006), the expansion of sea ice may be associated with a cooling triggered by a negative solar irradiance anomaly, which was amplified through a positive oceanic feedback mechanism. The cooling caused temporary relocation of deep-water formation sites in the Nordic Seas, which was accompanied by a distinct reduction in AMOC strength (Hall et al., 2004). The increase in sea-ice extent stratified the water column and hampered the deep-water formation, leading to additional cooling and more sea ice (Renssen et al., 2006)".

432: from ... to ...
Corrected.

433: Atlantic Water
Corrected.

437 to previous times
Corrected.

444: Several papers show that there was a cooling trend after c. 5 ka. However, there is also evidence for some warming in the last 2 ka (e.g., Sarnthein et al., 2003; Werner et al., 2013; Telesinski et al. 2014a,b; Zehnich et al. 2020). Is this expressed in your data? If not, can you speculate why?
Unfortunately, the resolution of the last 2000 years in our data is not high enough to see short term warming events, such as mentioned in e.g., Zehnich et al. 2020.

474: Syring et al., 2020b
Corrected.

501: Also Werner et al., 2013, 2016; Consolaro et al., 2018
Corrected.

523: Hillaire
Corrected.

539: started to resemble
Corrected.

542: Freshening is usually strongest near the sea surface and would thus increase (and not reduce) stratification...
As stated in the sentence, at the same time that the PW increased at the surface of the EGC, the warmer waters in subsurface levels transported by the RAC also decreased. With other words, the subsurface got also cooler, thus, reducing the stratification.

543: and to a (near) perennial
Corrected.

545: What is "possible sea-ice cover"?
Changed to "and possibly, sea-ice cover".

780: Sarnthein
Corrected.

Fig 1: Orange lettering is difficult to read on greenish background (insert map)
Thank you for your remark. We changed the color to slightly darker; however, we would like to keep the orange color because we would like to show to the readers that we speak about the same water masses on the insert map as the one marked orange on the overview map.

---

## Author Comment (AC2)

Referee 2

The paper by Pados-Dibattista et al presents new data from a remote site at the northeastern Greenland shelf aiming to reconstruct variability of climate and ocean conditions during the Holocene. The topic is within the scope of the journal and the presented data are new and results are very interesting. The paper is generally well written but has some aspects in the results and the discussion (see below) which need to be improved before the final publication. I recommend publication after major revisions given that authors can address the criticism detailed below.

We thank the reviewer for their valuable comments, which have been important for improving the manuscript and we have done our best to take into account. In a few instances, we have chosen a different solution to an issue pointed out by the reviewer; in those cases, we provide explanations to this under each comment below.

**Major comments:**

1. During the review I counted about 30 (!) abbreviations present in the paper and was often struggling with finding what those mean throughout the text. I suggest to the authors to make it easier for the reader, compile a list of all abbreviations used in the paper and present it in the beginning or alternatively at the end of the paper. This will facilitate finding them all in one place if needed.

The reviewer is correct that too many abbreviations are tiring. We have thus removed the abbreviations that were not repeated many times in the manuscript (PSW, AIW, NEW, NØIB, WSC). Moreover, as suggested, we added the list of remaining abbreviations (10) to the supplementary material.

2. Results section regarding foraminiferal analysis needs to be re-written starting with presenting a general information about foraminifera (this is present for planktonic but currently is missing for benthic ones). Such info shall include which benthic species were dominant (e.g. >10%), accessory (e.g. 5-10%) and rare (e.g. <5%). Then go into details regarding how many were calcareous vs how many were agglutinated and so on. The authors often write "the most abundant species" but it is unclear what they mean by those – dominant or accessory? This should be clarified.

The use of "dominant, "accessory", and "rare" was previously common when describing assemblages. However, most often this is omitted today, as we provide actual data on relative frequencies instead. However, we have now added these to the descriptions, where relevant.

3. The authors shall also add figures with absolute abundances because relative abundances can sometimes be misleading as they are usually not normalized per sample weight or sample volume. There are some sort of graphs present in the appendices, and although those look a bit different, they still have % instead of ind/g – if those are supposed to represent ind./g they shall be moved to the general text and be present alongside with % data instead of appendices.

The reviewer is correct that as relative frequencies are a closed sum, these calculations do not always tell if a % of a species increases just because other species decrease, and not because the first species is decreasing in abundance. However, relative abundances are in fact the data always used in interpretations and comparisons in palaeoceanographic studies, and we thus need to keep this figure. However, we have now added figures also on the foraminiferal species as individuals/g wet sediment to the supplementary material (Fig. A4 and A5); these figures will allow any reader test for actual changes in the individual species.

4. The authors should run some kind of multivariate statistics to strengthen their visual ecozone subdivision. A simple cluster analysis combined with PCA or Factor analysis would help here to verify if changes in the assemblages are significant enough to define ecozones. For instance, in results (p10, line 53) the authors define ecozone II based on appearance of *bulloides* in the record. This species, however, is present in very low abundances (<1%) and it is important to show with statistical analyses that its presence makes significant changes in the assemblages.

Some benthic foraminiferal species are highly sensitive to environmental changes, while others are more robust. Often the robust species dominate assemblages, while the rarer species may be the ones that are really important for palaeoceanographic and palaeoclimatic reconstructions. In these cases, multivariate statistics are not the best option for defining ecological changes, as it will never catch these more subtle but defining changes. This is the case here. Although it is understandable that the reviewer wonders about the significance that we place on *Pullenia bulloides*, its presence even in low numbers is highly important showing an un-usually high influx of Atlantic Water. Thus, it is here necessary to use this visual method for defining the zonation.

5. All ecozones are labelled in a strange manner e.g. "345-310 cm or 9.4-8.2 ka" – this should be changed to "310-345 cm or 8.2-9.4 ka" to make it more logical. Similar changes should be made all over in the text where authors refer to time periods or core intervals.

Thank you for your comment. However, it is common (see e.g., Moros et al., 2006; Jennings et al., 2011; Consolaro et al., 2018; Zehnich et al., 2020) and logical to describe and discuss a sediment core from the bottom (past) towards the top (present). Our text and thus the core depth and the age of the intervals follow this concept.

6. In addition to age, core depth should be added to all foram graphs to make it easier to follow the text and connection to the age model.

Good point. We added core depth to all of the foraminiferal graphs.

Also the width of the horizontal axis for the abundance data needs to be adjusted based on high (wide axis) or low (narrow axis) numbers – now all x-axis on graphs have the same width, but different range values, which makes it misleading for the reader

Thank you for your comment. The x axes have the same width because the ranges that they show (in some case 0-8, in some case 0-200) do not make it possible to have the same unit-length, and still present all data in one single figure. We believe that it is very important that all this data is shown in one single figure in order to compare them and understand easily the changes that happen through time. Moreover, this is a standard mode of presenting benthic foraminiferal data. In the figure caption the readers are warned about that "the x-axes have different scaling".

7. Section 5 (p11, lines 96-113) is way too long and contains a lot of text which belongs to the discussion. Here the authors shall only keep information about foram groupings they used for interpretation later– like suggested below. All remaining information including interpretation shall be moved to the discussion. A much shorter section 5 can include the following: Atlantic Water group includes x, y, z species (Refs); Chilled Atlantic Water group - x, y, z species (Refs); Arctic Water group ….S feylingi….Some agglutinated species have been described in connection with particular water masses (refs) and we use those to reconstruct…

We agree that this section would better fit into the discussion. We have now moved the paragraph to the discussion and, in addition, we have shortly listed the groups in the methods. Note that the move of the text to the discussion is not marked in yellow in the track changes version of the manuscript, as this would mask the other changes made this paragraph.

8. Section 6.2.1 (p13) shall be removed from the discussion as the authors say in the methods (p6, lines 190-19) that this section was not considered in the interpretation.

Thank you for your comment. We have now deleted the prior comment that the section is not discussed. We also removed the following sentence from the results: "This latter assemblage we interpret as reworked from older deposits.", as this is an interpretation not belonging in the Results section.

9. My last major point is about the discussion section, where the authors present their own data (section 6.2) separately from other studies (sections 6.3 and 6.4), which both shall be intertwined. So, the discussion needs to be rewritten.

Each paragraph in the discussion chapter 5.3 start with a short environmental interpretation of the data from our own core. It is necessary first to provide an environmental interpretation before discussing its significance. We chose to combine this environmental interpretation with the broader discussion in order to both avoid receptions and make the links clearer. However, we did keep the short interpretation in a separate subsection, in order to clearly separate, which part of the discussion is based on our new study, and which is based on comparison to previous studies.

The last two paragraphs of the discussion (5.4 and 5.5) place our results in an even broader context, but here we have now added more references to our own data.

**Minor comments:**

Abstract:

P1, Line 9: "We carried out benthic foraminiferal…" - This sentence jumps a bit abrupt from one topic to another. Start instead by saying "In order to reconstruct the variability of the East Greenland Current and general paleoceanographic conditions in the area during the Holocene, we carried out benthic foraminiferal assemblage, stable isotope- and sedimentological analyses etc…"

Corrected.

Introduction:

P1, Line 25: "According to model simulations, the Arctic Ocean may become seasonally ice-free as early as 2040-2050 (Stroeve 25 et al., 2012). However, despite the extreme societal and environmental relevance of this sea-ice reduction…" – please add a sentence or two explaining what these "extreme societal and environmental" implications are, and then introduce to the public the lack of knowledge on the topic.

We added a sentence.

Regional setting

P3, lines 83-85: Please remove abbreviations 79NG and ZI as those are only relevant of figure caption to Fig.1. Also please mark all "abbreviated" features (SS, ZI, 79NG, NQIB) on the Figure 1 with the arrows to better indicate their position on the map.

We removed the abbreviations, and we added Northeast Greenland Ice Stream with arrows showing towards the glacier outlets to the map.

P3, line 85: Please spell out 79NG and ZI.

Corrected.

Methods:
P5, lines 148-150: "A minimum of 300 benthic specimens were identified for each sample, except for four samples (10-11 cm, 15-16 cm, 30-31 cm, 195-196 cm). These four samples contained only between 242-296 specimens – I suggest authors removing the part about samples where 300 ind count was not reached. It is a well-known fact now that despite 300 ind/ sample is a standard procedure (e.g. Murray, 2006) counting individuals less than that can still produce statistically significant results (see e.g. Fatela & Taborda, 2002. Mar. Micropal. 45, 169-174).
We have now deleted the sentence "These data were, therefore, treated with some caution, but they were still included in the calculations for relative and absolute abundances". However, we believe that it is important to note, which samples did not reach the 300 specimens and also that the number was in fact not significantly below the 300 individuals.

P5, line 157: "..was restricted by the number of available, clearly identifiable tests…" – what does this mean? Not corroded and well preserved individuals? Please explain.
We mean with this sentence that only individuals were picked for stable isotope analysis that were with no doubt belonging to the species *Elphidium clavatum*. We omitted tests that raised any questions, in order to get the best possible results. However, in order to make it clear, we have added "without any corrosion or non-typical shapes" to the sentence.

Results:
P6, line 174: "The sediment of the lowest 40 cm of the core is much coarser…" -  it was mentioned in the methods that the authors did IRD analysis but that one is completely forgotten in the results and the discussion. Please add this information there.
Thank you for your comment. In the results, in chapter 4.1 (Core description) we write that our (low-resolution) IRD analysis revealed that between ca. 0-370 cm the sand fraction is on average 1.5%, without any significant changes that could be discussed. In the last 40 cm of the record, which probably contains reworked sediments and thus we omitted from further analysis, the sand fraction rises to 51%. These results are shown in Fig. 3. However, to make it more clear, we have now added that "in the rest of the core… the clay : silt : sand ratio does not show any significant changes".

P6, Figure 3: Please mark all available 14C dates close to the timescale.
Done.

P8, line 210: Please change the title to "Foraminiferal analysis" instead of "Foraminiferal content"
Corrected.

P8, line 211: Please change the title to "Redeposited core section 345-410 cm with Plio/Pleistocene Foraminifera" –Also I suggest the authors to swap the places for sections 4.4.1 and 4.4.2 and focus first on the core part which shows the main results. Then you can mention that the core base contains x,y,z forams, is likely redeposited, and is therefore excluded from interpretation.
Thank you for your comment. However, we believe that in the results section we cannot specify yet that this part of the core is redeposited, because the results serve as a plain description of the foraminiferal content, and adding "redeposited" to the title already presents an interpretation.

Moreover, the order of chapters follow the core description principle "from the bottom to the top i.e. oldest to youngest".

P8, line 219: Please change to "The Holocene core section 0-345 cm". Also consider swapping places with section 4.4.1 to present your most important findings first.
Please see the comment above regarding presentation of data and discussions from oldest to youngest.

P8, line 223 – p9, line 239: This section nee2ds to be re-written.
- Start with presenting foram concentrations as range (x-y) with an average z. Do this for both planktonics and benthics (calcareous and agglutinated separately). As it looks now, sometimes ranges are given but averages are missing, or the other way around. Please be consistent and present both all the time.

Thank you for your comment. We have now added concentration ranges and averages of species where relevant. However, we believe that it would make the results unnecessary long and not easily readable, if we would add ranges and averages to all species. Changes in abundances relative to the entire benthic assemblage and relative to the calcareous/agglutinate assemblage, supplemented with concentration changes (ind. /g sediment) are shown on Figs. 5, 6, A2, A3, A4, A5.
- Continue by telling which species were dominant, accessory and rare for each foram group (planktonics and benthics). Here, try to be consistent with spelling out species names fully every time they appear for the first time in a new section.

The use of "dominant, "accessory", and "rare" was previously common when describing assemblage. However, most often this is omitted today, as we provide actual data on relative frequencies instead. However, we have now added these to the descriptions, where relevant. Moreover, in order to keep the article shorter, we wrote out the full species names just at the first time they are mentioned in the manuscript. However, if the editor prefers us to always write out the species names in full, we are glad to do it so.

P8, line 230: "…from this point..." – which point? Please specify.
Changed to "from 210 cm until the top of the record".

P8, line 232: "..the most abundant.." – what does this mean? Dominant? Accessory? If so what range and averages this species has?
Please see our comment above about the categories "dominant", "accessory" and "rare". Moreover, the average abundances are stated in the same sentence: "the most abundant benthic agglutinated species are *Portatrochammina bipolaris*, followed by *Ammoglobogerina globigeriniformis*, representing on average 42 % and 16 % (respectively) of the benthic agglutinated assemblage, and 27.5 % and 10.5 % of the total benthic assemblage. They are both continuously present throughout the core and their relative abundances do not show strong fluctuations".

P8, line 233-234: "…representing on average 42 % and 16 % (respectively) of the benthic agglutinated assemblage, and 27.5 % and 10.5 % of the total benthic assemblage. " – information about their percentage within the agglutinated assemblage is irrelevant, please remove it and keep only % of the entire assemblage.
Thank you for your comment. However, we believe that it is important to show these species´ percentages within the agglutinated assemblage too, hence figure A3 in the supplement.

P8, line 236: "The most abundant.." – see comment above

Please see our comment above about the categories "dominant", "accessory" and "rare".

P8, line 237: Spell out *C. reniforme* and *E. clavatum*
Thank you for the comment, but they have been spelled out earlier (see chapter 4.4.1.), and in order to be consistent, we only spell out full species names once. Should the editor wish differently, we can of course easily add the full species names.

P9, lines 242-243, Figure 5 caption: "The depicted species were chosen in order to show changes in the environment." – this is vague and unclear. I assume those are changes in water masses such as…., if yes please specify.
Corrected and changed to "water masses".
Also please add core depth in cm to this graph, adjust axis width so it reflects the abundances visually and add a graph with respective changes for absolute abundances of those species.
We added core depth to the graph, however we cannot adjust the axis width; see our comment above, at major comment 6.

P9, lines 247-249: "..ecozones that were defined by visual interpretation of the species abundances" – please add here "within 0-345 cm core depth".
We added "in the dated section of the core (345-0 cm core depth)".

P9, line 250: please change the title to "Ecozone I. (310-345 cm; ca. 8.2-9.4 ka BP)". Do similar changes with swopping the ages and core depths so they appear in right order everywhere in the text.
Please see our comment about the order of age range and core depth above, at major comment 5.

P9, line 254: add "(unshown)" after the "agglutinated/calcareous ratio"
Done.

P9, line 256-258: "The benthic calcareous assemblage is dominated by *C. reniforme* (relative abundance on average 14 %), followed by *S. horvathi* (on average 12 %), *E. clavatum* (on average 11 %) and *C. neoteretis* (on average 9 %)." – where there any important accessory species?
In order to keep the results short and easily readable, we refer from naming all the species that are present in a certain interval. We mention only the species that are important regarding our environmental interpretation. For more details, please see Fig. 5 and 6 and the raw data.

P10, line 259: see my comment above regarding swopping the ages and core depths
Please see our comment about the order of age range and core depth above, at major comment 5.

P10, line 267:"..species *Adercotryma glomerata* significantly increases from 310 cm…" – based on figure 6 I cannot see any "significant" increase in *A glomerata* no matter how hard I try! So, I suggest to the authors to tone down this by removing word "significant" or to adjust the scale on *A glomerata* graph so it shows only 0-15% range with occasional peaks shown by axis break.
Corrected and removed "significantly".

P10: Figure 6 – please add core depth and adjust axis width so it reflects the abundances visually. Also add a graph with respective changes for absolute abundances of those species.

Thank you for your comment. We added the core depth to Fig. 5 and 6, however we cannot adjust the axis width; see our comment above, at major comment 6.

P10, line 273: see my comment above regarding swopping the ages and core depths
Please see our comment about the order of age range and core depth above, at major comment 5.

P10, lines 275: "rises dramatically" – an increase from 1 to 7 % is not a dramatic increase. Please tone this statement down or change to "comes back to higher numbers".
Corrected and removed "dramatically".

P10, lines 276-277: please add "Being present in abundances below 2% in ecozones I and II" to "Epistominella arctica increases…" Also remove "while the relative abundances of *A. glomerata* decrease" – as this is not visible from the Fig 6 as it looks now!
Thank you for your comment. It is clearly stated that *E. arctica* "increases (from 0.4 % (previous interval) to 0.8 %)"; but now we have added a short sentence pointing out that the species is still only present is low numbers. Moreover, we believe that the decrease of *A. glomerata* from ecozone II to ecozone III is visible from Fig. 6. In either case, the decrease is clearly present in the raw data and it is an important information to mention due to the ecological significance of these species.

P10, line 278: see my comment above regarding swopping the ages and core depths in the title.
Please see our comment about the order of age range and core depth above, at major comment 5.

P10, line 279: please remove word "drastic".  There are several instances in the record where calcareous benthics decrease but those are not mentioned.
Thank you for your comment. However, the benthic calcareous species continuously decrease from ecozone I to ecozone V, and we mentioned this at the beginning of the description of every interval. Ecozone II: "we can recognize a distinctive decrease in planktic and benthic calcareous foraminiferal concentration"; ecozone III: "The concentrations of planktic and benthic calcareous foraminifera continue to decrease…" ecozone IV: "The base of this ecozone is defined by the drastic decrease in benthic calcareous foraminiferal concentrations…"

P10, line 280: "…and increased relative abundances of *E. arctica* and *Stainforthia feylingi*." - Based on fig 5, *S. feylingi*has abundances quite similar to zones I and II but is not mentioned at all in the description of those.
Thank you for your comment. We mention certain species in the results when they show changes that could refer to significant environmental changes (also, relative to other species). In order to keep the results short and easily readable, we refer to mention the abundances of every species in every interval.

P11, line 282: "..agglutinated species *A. glomerata* shows a drastic peak at the beginning of this ecozone." – not just *A glomerata* but also *R. fusiformis* does the same and is worth mentioning here. Please add.
Added *R. fusiformis*.

P11, line 283: "..*Spiroplectammina biformis* increase significantly compared to the previous interval." – please consider changing to "*Spiroplectammina biformis* starts to increase as compared to the previous ecozones.

Corrected.

P11, line 284: see my comment above regarding swopping the ages and core depths in the title.

Please see our comment about the order of age range and core depth above, at major comment 5.

P 11, line 288: "*Saccamina difflugiformis* shows a steep rise unique to this interval" – this species again has been present in other ecozones as well but is completely ignored in the description of those. Why?

Thank you for your comment. We mention certain species in the results when they show changes that could refer to significant environmental changes (also, relative to other species). In order to keep the Results short and easily readable, we refer to mention the abundances of every species in every interval.

P11, lines 296-313: This section needs to be completely rewritten and moved to the methods (see suggestion below). Note that all information containing interpretation needs to be moved to the discussion. You may want to keep the following information (copy pasted from the MS) and move that to the Methods rather than keeping this in the results:

"In order to be able to describe the changes in water masses over time on the NE Greenland shelf, we place selected benthic calcareous foraminifera species into groupings that are based on environmental preferences of the species (Table A2 in appendices). *The Atlantic Water group* includes *C. neoteretis* and *P. bulloides* (e.g., Mackensen and Hald, 1988; Seidenkrantz 1995; Rytter et al., 2002; Jennings et al., 2004; 300 Jennings et al., 2011; Cage et al., 2021). *The chilled Atlantic Water group* includes *I. norcrossi* and *M. barleeanum* (e.g., Slubowska-Woldengen et al., 2007; Perner et al., 2011; Perner et al., 2015; Cage et al., 2021). *The Arctic Water group* includes *S. horvathi* and *E. arctica* (e.g., Green, 1960; Lagoe, 1979; Wollenburg and Mackensen, 305 1998; Jennings et al, 2020). *Stainforthia feylingi* is used here as a sea-ice edge indicator species that tolerates unstable conditions (Knudsen and Seidenkrantz, 1994; Seidenkrantz, 2013); its increase may refer to the location of a sea-ice margin at the study site. Moreover, we use in the interpretation the abundances of the agglutinated species *A. glomerata*, *T. earlandi*, *T. torquata*, *S. biformis* and *S. difflugiformis*, as those have been linked to specific water masses in the Arctic (e.g., Hald and Korsun, 1997; Jennings and Helgadottir, 1994; Korsun and Hald, 2000; Lloyd 2006; Perner et al., 2012; Perner et al., 2015, Wangner et al., 2018).

We agree that this section would better fit into the discussion. We have now moved the paragraph to the discussion. In addition, we have shortly listed the groups in the methods.

Discussion:

P12, lines 324-325: Please change: "the temperature profile" to "CTD profile", "not exceeding" to "below" and "closer in character to that" to "closer in character to the water"

Corrected.

P13, line 326: please change "the site" to "our study site" and "but may also be due to wind-driven upwelling" to "but may also be present due to wind-driven upwelling

Corrected.

P13, lines 332-338: This section needs to be removed – see my major comment 7.
Please see our comment above, at major comment 8.

P13, line 339: please change "9.4-8.2" to 8.2-9.4"
Please see our comment about the order of age range and core depth above, at major comment 5.

P13, line 340: Before jumping into the discussion, you need to lead to it first based on what your data show. E.g. start with saying something like " The AMS 14C dating places this core interval into the early Holocene, which based on our data and previous studies (Refs) is suggested to be dominated by colder climate conditions over the study site. This cold interval was characterized by…."
Deleted "cold".

P13, line 342: "Atlantic water indicator species" – please add "such as" and list those. Also change "points to" to "suggests".
Corrected and added.

P13, line 349: please change "was also characterised" to "was also likely characterised"
Corrected.

P13, line 350: please add "and" between "foraminifera" and "the presence of E. arctica"
Thank you for your comment, however there was already an "and" at the end of the listing: "…foraminifera, the presence of *E. arctica*, which thrives in high-productivity environments (Wollenburg and Kuhnt, 2000) **and** a pronounced peak in Ca/Fe ratio…". However, we have now divided up the sentence to prevent any misunderstanding.

P14, line 370: please change "8.2-6.2" to "6.2-8.2'"
Please see our comment about the order of age range and core depth above, at major comment 5.

P14, lines 373-375: "*C. neoteretis* and *A. glomerata* had their highest relative abundances during this interval…and *P. bulloides* appeared in the record after a long absence around 8 ka BP (Figs. 5 and 6), suggesting highly stable bottom waters (Rytter et al., 2002)." " - please spell out *C. neoteretis* fully or change to "Species C. neoteretis" (this applies to all species names at the start of a new sentence, as it is not correct to start a new sentence with an abbreviation).
Corrected.
Also abundances of *A. glomerata* in zone II are not much higher than in zone 5, associated with Neoglaciation, so please change the statement about its highest abundances.
Thank you for your comment. The reviewer is correct that the abundances of *A. glomerata* in ecozone II are not much higher than in ecozone V, however, this does not change the fact that *A. glomerata* still has its highest abundances in ecozone II.
As for *P. bulloides* – see my comment regarding its abundances <1%. The authors should discuss this and run a multivariate statistical analysis to see if this assemblage change is significant or not and add a discussion regarding this in the text.
Please see our comment about the statistics above, at major comment 4.

P14, line 388: "..on the ME and SE Greenland shelf…" – what does abbreviation ME stands for? Please explain.

Corrected and added Middle East and South East.

P14, line 391: please change "6.2-4.2" to "4.2-6.2" in the title

Please see our comment about the order of age range and core depth above, at major comment 5.

P15, line 407: please change "4.2-3.2" to "3.2-4.2" in the title

Please see our comment about the order of age range and core depth above, at major comment 5.

P15, line 431: please change "3.2-0.3" to "0.3-3.2"

Please see our comment about the order of age range and core depth above, at major comment 5.

Sections 6.3 and 6.4: This sections solely focus on findings of the others instead of putting own data into perspective of other studies. I suggest to move these sections to the introduction where they more naturally belong in their present form, otherwise the authors shall make an effort in answering the question how their own data fit into other studies they give an overview to herein. Go back to your aim. What are your data telling you about dynamics of the water masses you aimed to reconstruct? E.g. was EGC weaker or stronger? What does this mean for AMOC and climate in the Arctic (based on other studies)? What are future implications of those changes (again based on other studies)? As for now the authors treat their own data separately from other studies and this makes the last two sections look a bit off place giving a feeling that those shall be moved to the introduction or included in a review paper on the topic.

The last two paragraphs of the discussion (5.4 and 5.5) place our results in a broader context. However, here we have now added more references to our own data to emphasize how our data fits in the context.

Conclusions:

This section contains too much information in its present form and is difficult to read due to its bullet-point structure. I suggest condensing conclusions, to present the most essential findings, again linking back to the hypothesis, aim and motivation of the study.

Thank you for your comment. However, this is a matter of personal writing style and we think that exactly the bullet-point structure makes the conclusions easily understandable and quick to read-through. Unfortunately, we cannot reduce the information shown here, as we wrote only one – max. two sentences about each interval/topic.

---

## Author Response (AR2)

Dear Editor,

Thank you very much for your comments and suggestions. We have now made the changes and added references in all places you required it. Please see below the answers to your comments.

**Page 2, line 60**: We have deleted "West Spitzbergen Current", however, we think that it is necessary to keep "RAC", as we use this abbreviation several times in the manuscript, in some cases also two times in one sentence. However, if the Editor wishes so, we will write out Return Atlantic Current at every occasion throughout the paper.

**Page 3:**
Where do you refer to when you say northern North Atlantic? Her it sounds like you refer to the impact of fresh water in teh the North Atlantic (south of teh ridge) - but in teh sentence above it sounds like the northern North Atlantic is used for the eastern Nordic Seas. Be precice to avoid confusion.
Thank you for your comment; we have now changed the location names to more precise terms.

Unclear; what do you mean by dynamics of the Arctic Ocean? Add reference.
Noted. We have now changed it to "ocean-atmosphere dynamics", have added an example of what we mean by it, and have added two references to the sentence.

The part related to NAO/AO would benefit from a rewriting:
Complex - rather use mode of variability
redistribution of air masses - be more concrete; these modes of variability is defined with respect to the fluctuations in atmospheric preassure at sea level (for NAO SLP between the Azores high and Icelandic low).
Positive NAO - stronger southwesterlies rather than straight westerlies (that are more the case during negative NAO phases)
As suggested, we have changed "complex" to "mode of variability", and we added information about the fluctuations of atmospheric pressure at sea level. We have also exchanged "westerlies" to "south-westerlies".

**Page 10**
Confusing; in the text the dark blue species are linked to permanent sea ice conditions. Here they are linked to Arctic water. Please clarity. Arctic water is not permanently sea ice covered.
You are correct; the explanation in the figure caption was too simplified. We have now added "often living beneath perennial to near-perennial sea-ice" to the description of the Arctic Water species. Further details, i.e. that *S. horvathi* is linked to perennial sea ice, while *E. arctica* is rather found in connection to open water areas within the sea ice we only explain in the actual text, not the figure caption in order to not repeat text and make the caption too long and complicated. However, we have expanded this explanation and the comparison to *S. feylingi* in the text. We also added specified that the light blue species (*Stainforthia feylingi*) is linked to the sea-ice edge in the figure caption.

**Page 15**
The fresh melt water will normally forma lid at the surface. By what mechanism do you get an impact of the fresh meltwater on the d18O values of the bottom water? Please specify.
Thank you for your comment. We added the information to the sentence that the melt water might sink with melt water plumes to the bottom.

What do you mean by heavy sea ice cover? Permanent sea ice?
In the next paragraph you pulll support for this interpretation from the Zehnich et al., 2020 paper; however, whiel they do argue for more sea ice at this time than earlier, this interpretations is linked to the contemporary low phytoplancton growth. And if you look at their results in respect of sea ice coverage it transfer from reduced/variable to marginal/seasonal during this interval, not permanent. If you are closer to marginal/seasonal than permanent sea ice, your high productivity is more logical. Clarify your interpretation and your discussion, so that it is clear in the end what your prefered interpretation is, and why, not only based on your records but seen in context of available regional information.
In addition to the records already mentioned, Maffezzoli et al., 2021 QSR present a new sea ice information from the RECAP ice core at Renland.

Unfortunately, with our methods we are not able to produce a quantitative sea-ice reconstruction. Thus we cannot with certainty differentiate between perennial and extensive (near-perennial) sea-ice cover, thus, we prefer not to name it "perennial" or "near-perennial" and just use the broader term "extensive". However, as suggested, we have deleted the part that states high bioproductivity referred by Zehnich et al., 2020.

**Page 16**
Pure speculation; do you have any evidence/support for this statement? If not I would delete this sentence.
We have reformulated the sentence and supported the statement with reference to previous studies.

Use the information from other studies actively as part of the discussion above. E.g. Our interpretation of a stronger influence of AEW at our site is supported by ...
Same as above and check throughout - make sure to link the information from existing studies to the interpretation of your data. Here you provide statements on what they find, but do not discuss.
We have reformulated several sentences in the discussion to link our interpretation better to existing studies.

**Page 17**
How do you get reduced stratification by freshening? Normally a fresher water mass will have a lower density and hence enhance the stratification.
We see that this could seem strange. We have thus added reduction of "warmer waters" to the sentence, in order to make it clear that parallel to the colder surface waters the subsurface waters also got cooler with the reduction of RAC.

**Page 18**
This section reads more like a review of litterature than a discussion of your results in context of knowledge from the litterature.
I tenk to agree with the referees that it would strengthen the paper if you integrate the information of section 5.4 and 5.5 at the relevant places in the previous sections.
Thank you for your comment. We have now integrated chapter 5.4 into chapter 5.3. However, we would like to keep chapter 5.5 (now 5.4) separated, as this part describes a mechanism that stretches over the whole time period discussed and while the previous chapters of the discussion primarily discuss the general development, this chapter is more focused on processes.

Resolution is also a key factor here - its a very short event so to be able to detect it you need better resolution than what you have in your foraminifera data.
Thank you for your comment, however it is already mentioned earlier, in chapter 5.3.2: "It is not recognizable in the foraminiferal assemblage changes/stable isotope results, although the lower temporal resolution of the latter may also not allow us to identify any changes." As we have now integrated the part that you are referring to into chapter 5.3, we think that it is not necessary to repeat this statement.

Add reference.
We have deleted this sentence, because after the integration of the two chapters it became redundant.

Delete.
Or clarify the relation to your results / how the change in the Irminger Current is physically conected to the suggested change in the EGC.
The first part of the sentence may be kept if you rephrase it, relating the downstream warming to the increased influence of Atlantic Water at your site.
Thank you for your comment. We have now reformulated the sentence to make it more clear the connection between the EGC and the northern flow of Atlantic water.

**Page 19**
how do you know that the fresh water forcing commes from the Arctic Ocean?
We see that this was unclear. We have now added references to this sentence.

Specify the changes and add reference for the covariance statement.

The term HTM is used in many different ways and the timng may change depending on definition used and where you are, please add information about when this happened.
Same bellow with mid Holocene - you define it previously, but its easuer to follow for the reader if you use ages, and its more specific.
Thank you for your comment. The sentence you are referring to, was meant as an introduction to this section; the changes are specified following this sentence. However, now we have reformulated a bit and added a reference to make our intention clear. We have also added ages to HTM and mid Holocene.

Depending on an assumption that the fraction of AW entering the Arctic Ocean and the RAC staying constant.
We have now added this information to the sentence.

The NAO/AO statement comes out of the blue. It is not clear from the above why this coolingis associated with a transition from a positive to a negative NAO. Please rephrase to clarify.
We have rewritten this part to make the statement more clear.

**Page 20**
Andreas Born have several newer papers investigating the dynamics of the SPG
We have added reference to a paper from Born.

---

## Author Response (AR3)

Dear Editor,

We hereby follow your request and have deleted the sentence in question. Moreover, we have added two references to the previous sentence that describe locations along the pathway of northern flow of AW.

Kind regards,
Teodora Pados-Dibattista